# Source-to-sink coupling and spatiotemporal evolution during multiphase rifting: The Jurassic-Cretaceous Lishu fault depression, Southeastern Songliao Basin

Ke Wang [1,2], Yong Zhou[1,2]*, Jingchi Yan[1,2‡], Yuejie Zhang[1,2‡]

**1** College of Geosciences, China University of Petroleum (Beijing), Beijing, China, **2** National Key Laboratory of Petroleum Resources and Engineering, China University of Petroleum (Beijing), Beijing, China

☺ These authors contributed equally to this work.
‡ These authors also contributed equally to this work.
* Wangke253@163.com

## Abstract

Source-to-sink (S2S) systems exert a fundamental control on sediment dispersal and reservoir development in multiphase rift basins; however, their coupling mechanisms and spatiotemporal evolution remain poorly constrained. The Lishu Fault Depression, located in the southeastern Songliao Basin of Northeast Asia, represents a typical example of a multiphase rift lacustrine basin. Nevertheless, incomplete understanding of its S2S dynamics has hindered effective hydrocarbon exploration and development. In this study, sequence stratigraphy, sedimentology, and S2S system theory are integrated with seismic, drilling, and well-log data to construct a high-resolution sequence stratigraphic framework and to quantitatively characterize sediment provenance, transport pathways, and depositional systems. On this basis, paleo- S2S systems are reconstructed and systematically classified. The results indicate that seven sequence boundaries can be identified within the Jurassic-Cretaceous succession of the Lishu Fault Depression, delineating six third-order sequence stratigraphic units (SQ1-SQ6). Analysis of boundary fault activity reveals that the Sangshutai Fault exhibits higher displacement rates in its central segment and progressively weaker activity toward both ends, with peak fault activity occurring during the late stage of intense rifting. During this rifting phase, sediment supply was dominated by multiple provenance systems, while sediment-routing pathways primarily developed as fault-controlled valleys, structural transfer zones, and parallel fault-step zones. These pathways exerted a strong control on the spatial distribution of nearshore subaqueous fans, fan deltas, braided river deltas, and lacustrine deposits. From the early rifting stage through the rift atrophy stage, the basin evolved from a tectonic framework characterized by alternating uplifts and depressions into a unified lacustrine basin, and subsequently into a shallow, laterally extensive lake system. During the initial rifting stage, steep-slope

**Data availability statement:** All relevant data are within the manuscript and its Supporting information files.

**Funding:** This research was funded by National Natural Science Foundation of China project "Research on the Mechanism of the Role of Saline Substances in Saline Lake Basins on the Evolution of Diage-netic Fluids and Differential Diagenetic Responses in Deep Tight Reservoirs" (Project No.: 42472178).

**Competing interests:** The authors have declared that no competing interests exist.

nearshore subaqueous fans and slope–fan delta systems developed. In the early phase of intense rifting, dominant patterns included steep slope–transfer zone–fan delta, axial slope–fan delta, and eastern gentle slope parallel fault-step zone systems. In the late phase, axial drainage systems prevailed, forming axial slope–braided river delta patterns. During the late rifting stage, gentle slope–braided river delta source-to-sink systems became dominant. Quantitative analyses further demonstrate that, during the intense rifting stage, the scale of gravity-flow-dominated nearshore sub-aqueous fans and traction-current-dominated deltaic deposits is positively correlated with the width, depth, and cross-sectional area of sediment-routing valleys. These results elucidate the dynamic coupling between tectonic processes and S2S system evolution in multiphase rift basins, and provide a robust geological basis for predicting sand body distribution and guiding deep hydrocarbon exploration in the Lishu Fault Depression and analogous rift basin settings.

## Introduction

In recent years, source-to-sink (S2S) systems have become a widely adopted frame-work for linking geomorphic evolution, sediment routing, and depositional architecture in sedimentary basins. By integrating sediment provenance, transport pathways, and depositional systems, the S2S approach provides an effective means of recon-structing sediment dispersal processes and elucidating their tectonic and geomorphic controls, particularly in tectonically active basins [1–6]. In contrast to passive-margin or ocean-continent transition basins, which are commonly characterized by large and persistent axial river systems, continental rift basins typically display complex, multi-source sediment supply patterns [7–9]. These basins are strongly influenced by rapid fault growth, shifting accommodation zones, and variable uplift-subsidence regimes, resulting in pronounced spatial and temporal heterogeneity in sediment routing systems. Recent advances in S2S modeling indicate that episodic rifting exerts a first-order control on drainage reorganization, transport pathway geometry, and dep-ositional system distribution through processes such as fault linkage, displacement transfer, and depocenter migration [10–14]. Over the past decade, quantitative S2S models have increasingly integrated fault growth histories, drainage reorganization, sediment supply, and depositional architecture in continental rift basins. The com-bined application of forward stratigraphic modeling, high-resolution seismic data, well-log interpretation, and provenance analysis has enabled detailed reconstruction of sediment dispersal patterns, transport pathways, and depositional systems. These studies demonstrate how multiphase faulting, accommodation variability, and paleo-geomorphology govern sediment routing and basin-scale stratigraphic architecture, thereby improving predictions of sand-body distribution, reservoir potential, and S2S coupling processes during rift evolution [15–22]. The Lishu Fault Depression, located in the southeastern Songliao Basin of Northeast China, represents a typical multi-phase rift lacustrine basin. It is characterized by nearshore subaqueous fans devel-oped along steep fault-controlled slopes and extensive deltaic systems distributed

across gentler slopes [23,24]. Despite its significance, a systematic S2S coupling model for the rifting stage of this depression has yet to be established. Utilizing newly acquired three-dimensional seismic data in combination with drilling and well-log information, this study examines the Jurassic-Cretaceous rifting stage of the Lishu Fault Depression from a S2S perspective. The primary objectives are to characterize sediment provenance, transport systems, and depositional architectures; to reconstruct and classify S2S systems associated with multiphase rifting; and to establish a quantitative coupling model that links fault activity, sediment routing, and depositional responses. The results provide new insights into S2S system evolution in multiphase rift basins and offer a robust geological basis for predicting sand-body distribution and identifying favorable reservoir zones in the Lishu Fault Depression and comparable continental rift settings.

## Regional geological background

The Songliao Basin in northeastern China is one of the largest Meso-Cenozoic continental petroliferous basins in the region [25–27]. Shaped by multiple phases of tectonic activity, the basin exhibits a well-defined structural framework characterized by east-west zonation and north-south segmentation. This framework comprises six major tectonic units: the Central Depression Belt, Eastern Uplift Belt, Northern Slope Belt, Southeastern Uplift Belt, Western Slope Belt, and Southwestern Uplift Belt [28,29]. The Lishu Fault Depression is located in the southeastern sector of the Southeastern Uplift Belt (Fig 1a) and covers an area of approximately 2,346 km². It represents a relatively large and deep half-graben system formed through fault-depression superposition since the Late Jurassic. Structurally, the depression is characterized by a "west-bounding fault and eastward onlap" configuration and exhibits a framework described as "three sags, two slopes, and one uplift," with stratigraphic thickness progressively decreasing from west to east (Fig 1b).

Based on fault geometry and deformation characteristics, the Lishu Fault Depression can be subdivided into six structural zones: the Sangshutai Sub-sag, Shuanglong Sub-sag, Sujiatun Sub-sag, Northern Slope Belt, Central Uplift Belt, and Southeastern Slope Belt.

The stratigraphic succession of the Lishu Fault Depression is well preserved and comprises, from youngest to oldest, Quaternary, Neogene, Paleogene, and Cretaceous strata. The basin basement is composed of Carboniferous-Permian metamorphic rocks. Strata deposited during the fault-depression stage attain an average cumulative thickness exceeding 5,000 m, whereas deposits formed during the subsequent thermal subsidence (depression) stage are approximately 2,000 m thick. Within the Lower Cretaceous succession, the Huoshiling, Shahezi, Yingcheng, and Denglouku formations are developed in ascending stratigraphic order [30]. The Huoshiling Formation is characterized by volcanic rocks interbedded with clastic units, with volcanic facies preferentially developed over structural highs and thick clastic successions accumulating within localized, fault-controlled depocenters. The overlying Shahezi Formation unconformably overlies the Huoshiling Formation and is dominated by dark grey mudstone interbedded with thick intervals of grey, fine-grained sandstone and subordinate conglomeratic sandstone. This formation has a total thickness of approximately 600–800 m (Figs 1b and 2). The lower part of the Yingcheng Formation is composed mainly of thick mudstone, whereas its middle to upper intervals contain two to three sets of upward-coarsening sand-mud cycles. Along the basin margins, the Yingcheng Formation is commonly truncated, resulting in incomplete preservation; its thickness ranges from 450 to 1,000 m. The overlying Denglouku Formation disconformably or angularly unconformably overlies the Yingcheng Formation and is dominated by thick grey conglomeratic sandstone interbedded with brown to grey-green mudstone. The thickness of the Denglouku Formation varies between 100 and 500 m [31,32] (Fig 2).

## Materials and methods

### Materials

High-quality three-dimensional seismic reflection data covering approximately 2,300 km² of the Lishu Fault Depression, together with data from more than 50 exploration wells constrained by vertical seismic profile measurements, were

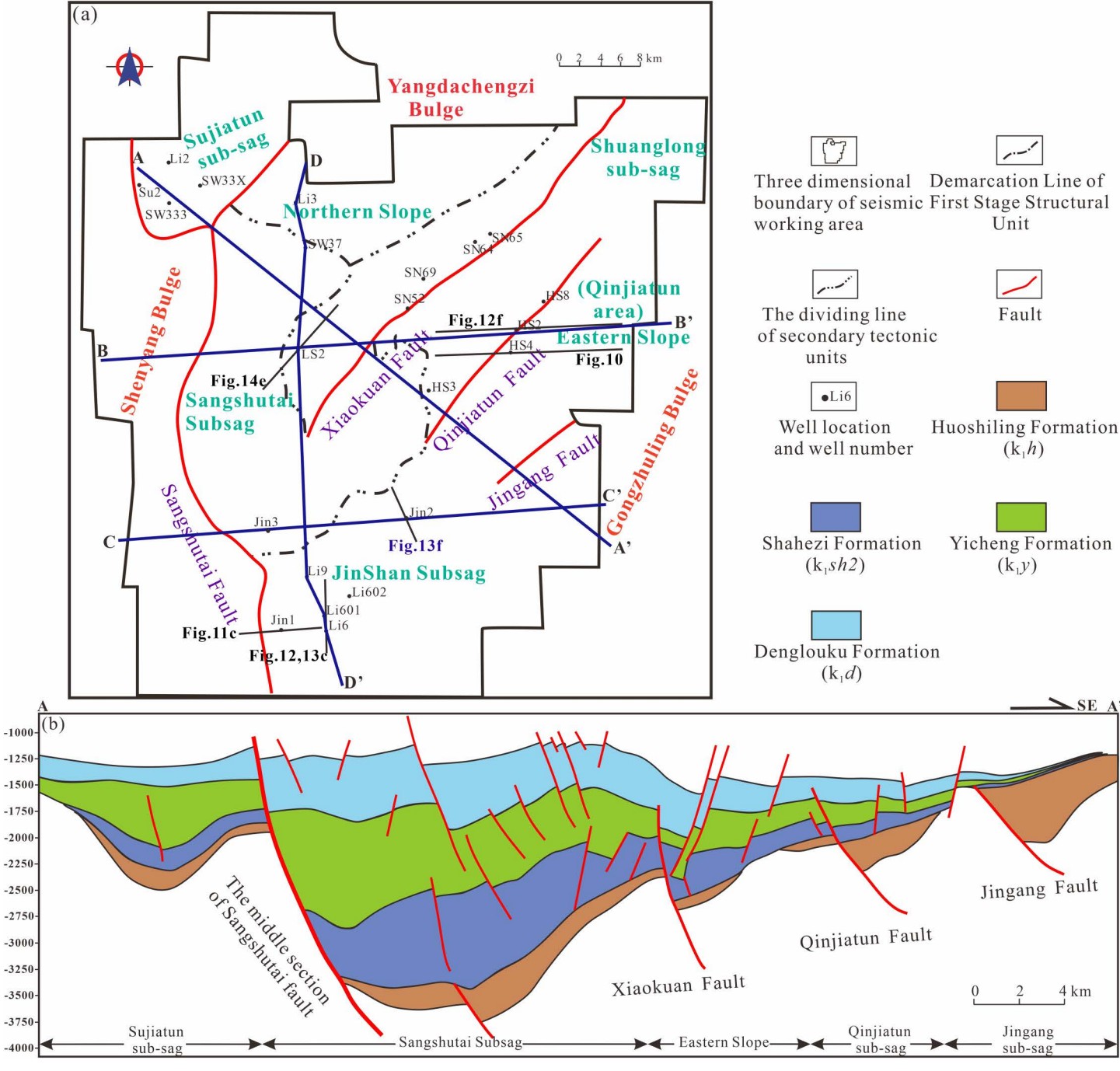

**Fig 1. Tectonic location and stratigraphic division characteristics of the Lishu Fault Depression (a) Tectonic unit division and regional overview of the Lishu Fault Depression; (b) Stratigraphic development characteristics of the Lishu Fault Depression.** (Note: Fig 1a was generated using Petrel 2022 and subsequently imported into CorelDRAW for final editing and illustration).

**Fig 2. Stratigraphic evolution histogram of the Lishu Fault Depression [30].**

provided by Sinopec Northeast Oil Company. The seismic dataset enables detailed structural and stratigraphic interpretation throughout the depression, whereas well logs and core data provide critical lithological, porosity, and stratigraphic calibration. Collectively, these datasets support reliable time-depth conversion and facilitate the quantitative reconstruction of S2S systems.

## Methods

**Quantitative analysis of fault activity rate.** The faults analyzed in this study are synsedimentary structures formed under extensional tectonic regimes. These faults display systematic thickness variations between hanging-wall and footwall strata, reflecting differential accommodation generated during periods of active faulting. When fault movement persists during sediment deposition, increased accommodation space develops on the hanging wall, resulting in strata

that are significantly thicker than those on the footwall. To quantify temporal variations in fault activity, the fault activity rate was calculated as follows:

Fault activity rate = (hanging-wall thickness – footwall thickness)/ duration of deposition.

Fault throw was quantified through interpretation of seismic profiles oriented perpendicular to the fault strike. Stratigraphic ages were constrained using regional and adjacent sedimentary age data to ensure the consistency and reliability of the temporal framework adopted in this study [31].

**Analysis of source system, transport system, and depositional system.**

(1) Paleogeomorphology Reconstruction: Reconstruction of paleolandforms in source areas aims to restore the original tectonic framework of drainage systems and provides a fundamental basis for elucidating the developmental characteristics and spatial organization of S2S systems [33–35]. The paleogeomorphic reconstruction procedure adopted in this study consists of four principal steps.1) Residual landform reconstruction. Geological interpretations are first integrated with drilling and stratigraphic data. Stratigraphic boundaries identified on key seismic profiles are used to delineate the lateral distribution of residual formation thickness. By correlating the upper and lower stratigraphic surfaces within the seismic survey area, the present-day residual thickness is calculated as the difference between the top and bottom surfaces. 2) Differential compaction correction. Porosity-depth relationships derived from drilling and well-log data are employed to establish a porosity-depth function. Based on the principle of constant grain volume, compaction-related burial loss is quantified and applied to correct the original formation thickness. 3) Restoration of denudation thickness. The stratigraphic extension trend method is applied to reconstruct pre-erosional stratigraphic geometry and distribution. By extrapolating the trends of preserved strata into denuded areas, the difference between the reconstructed and preserved thicknesses is calculated to determine denudation thickness. 4) Three-dimensional paleogeomorphic visualization. After applying compaction corrections and restoring total material removal, the denudation thickness is combined with the residual formation thickness to obtain the total stratigraphic volume for the target interval. Three-dimensional rendering is subsequently performed using geological modeling software to generate the paleogeomorphology of the corresponding period.

(2) Source Analysis: Sandstone petrography (QFL plots) constrains tectonic setting of source areas. Heavy-mineral assemblages refine source discrimination [36,37]. Core and well-log data are used to establish lateral correlations and to evaluate potential sediment mixing or diagenetic alteration. The observed compositional variations are interpreted as reflecting genuine changes in sediment supply and are consistent with the stratigraphic architecture.

(3) Identification of Transport Channels: Seismic profiles perpendicular and parallel to inferred sediment pathways are analyzed. Valley-type channels, transfer zones, and fault-plane interactions are identified based on their geometric attributes and seismic reflection characteristics. Well logs validate depositional contacts and channel extents. Differential activity at fault tips and specific fault interactions help classify sediment routing types.

**Statistics of valley parameters and sedimentary fan scale.** Valley geometric parameters, including width, depth, and cross-sectional area, as well as upstream drainage areas, were extracted from seismic and well-log data.

Areal extent of nearshore subaqueous fans and deltas is measured using Petrel 2022.

Scatter plots in Excel establish quantitative relationships between S2S elements.

To minimize uncertainty associated with overlapping depositional bodies, only depositional systems formed from the initial rifting stage through the late intensive rifting stage were included in the analysis.

Potential sources of error, including depth-conversion inaccuracies and uncertainties in seismic-attribute measurements, were explicitly considered.

## Results

### Sequence stratigraphic framework division

In this study, the stratigraphic interval from the Huoshiling Formation to the Yingcheng Formation in the Lishu Fault Depression was subdivided into the initial rifting stage and the strong rifting stage, within which SQ1-SQ6 were identified (Fig 3). Six sequence boundaries were recognized based on characteristic seismic reflection features. Boundary T5, corresponding to the basement unconformity, is marked by widespread truncation and onlap. Boundary T42, representing the top of the Huoshiling Formation, is characterized by medium- to strong-amplitude reflections with pronounced lateral variability and down-truncation and onlap relationships. Boundaries T41a (top of the first member of the Shahezi Formation) and T41 (top of the Shahezi Formation) display clear contrasts between overlying and underlying seismic packages, largely conformable contacts, and prominent onlap at the basin margins. Boundary T4a, marking the top of the second member of the Yingcheng Formation, exhibits similar seismic characteristics, including strong reflection contrasts and marginal onlap. Boundary T4, corresponding to the top of the Yingcheng Formation, is a regional unconformity characterized by strong lower reflections, weaker upper reflections, and widespread onlap. Boundary T3, representing the top

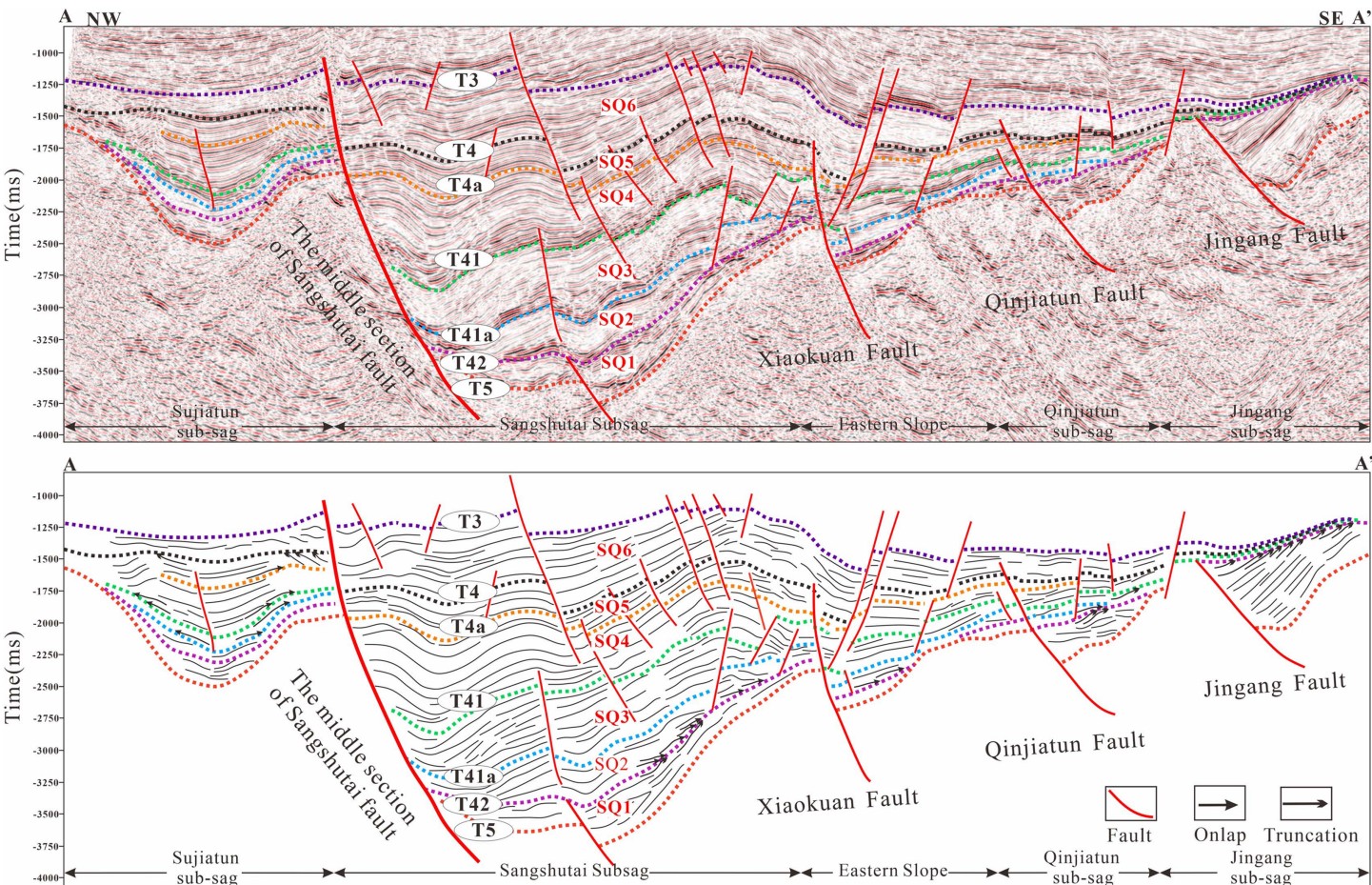

**Fig 3. Interpretation of NW-SW seismic profile and sequence stratigraphic framework in the Lishu Fault Depression (The location of the seismic profile is shown in Fig 1a).**

of the Denglouku Formation, is also a regional unconformity, distinguished by distinct reflection differences between the overlying and underlying strata (Figs 3 and 4). From a chronostratigraphic perspective, SQ1 corresponds to the initial rifting stage represented by the Huoshiling Formation, SQ2-SQ3 to the early strong rifting stage of the Shahezi Formation, SQ4-SQ5 to the late strong rifting stage of the Yingcheng Formation, and SQ6 to the rifting atrophy stage represented by the Denglouku Formation. SQ1 is characterized by pronounced lateral thickness variations, with extensive erosion along basin margins and sediment accumulation largely confined to early, isolated small fault depressions. SQ2, deposited during the early phase of rapid rifting, records the progressive amalgamation of initially isolated fault depressions into a unified west-faulted, east-onlapping half-graben system. During this stage, relatively weak basement fault activity resulted in thin and spatially restricted sedimentary successions, primarily developed within the Sangshutai Sag. SQ3 documents

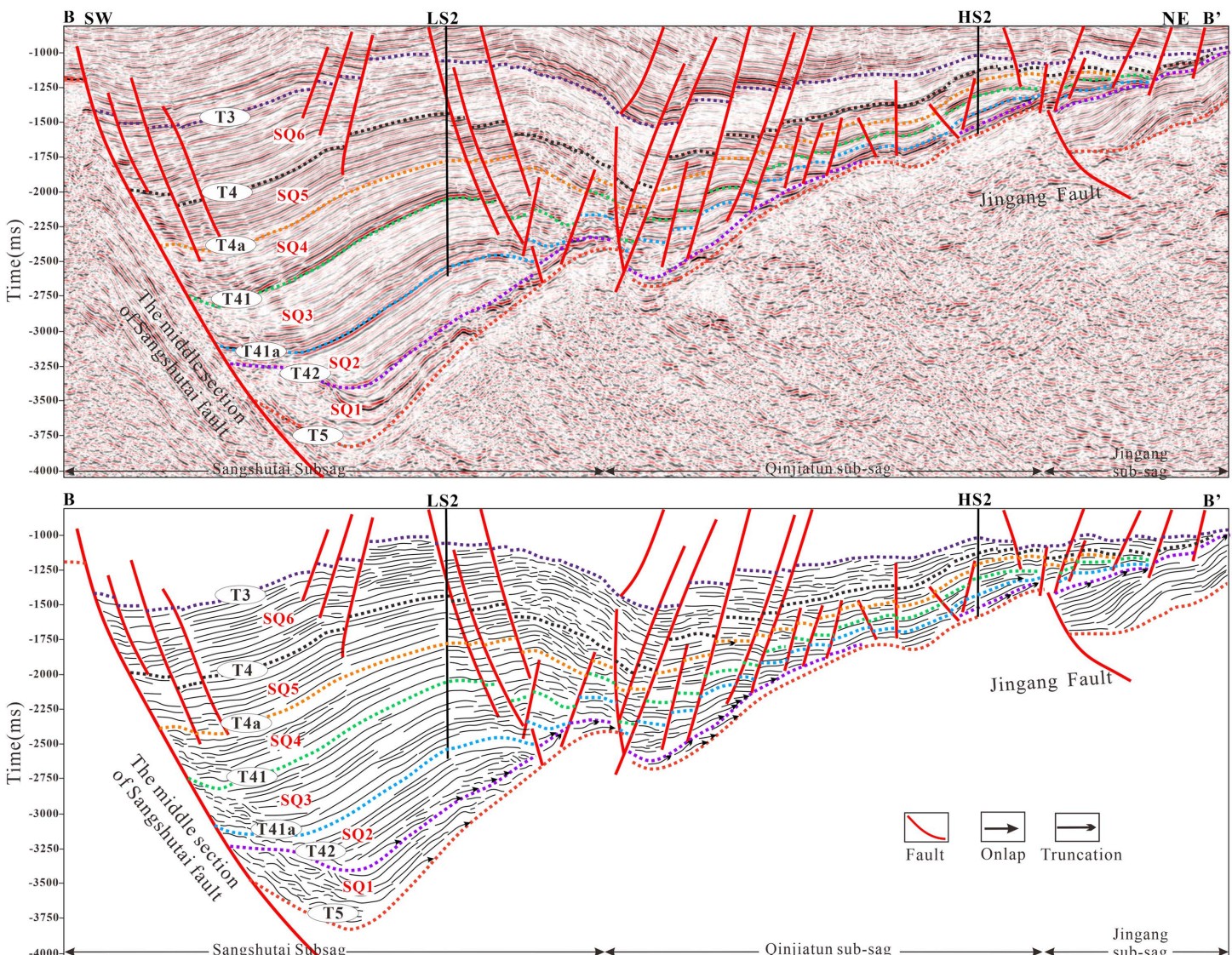

**Fig 4. Interpretation of SW-NE seismic profile and sequence stratigraphic framework in the Lishu Fault Depression (The location of the seismic profile is shown in Fig 1a).**

a transgressive expansion of the lacustrine system, characterized by the development of deep- to semi-deep-lake mudstones in the basin center and gradual thinning toward the north and east, reflecting erosion along the Northern Slope Belt. SQ4 and SQ5, corresponding to deposition of the Yingcheng Formation, record sedimentation and subsidence concentrated in the Sujiatun Sub-sag and Sangshutai Sag, where maximum thicknesses were attained. Strata thin progressively eastward and northward, and sedimentary slopes developed along the southeastern, northern, and western basin margins. During the late highstand stage, tectonic uplift led to substantial erosion of the Northern and Southeastern Slope Belts. SQ6 reflects basin-wide sedimentation that progressively onlapped onto uplifted areas, ultimately resulting in the formation of a unified and laterally extensive depression by the end of Denglouku Formation deposition [38,39].

## Fault activity analysis

The Sangshutai Fault, which constitutes the principal boundary structure of the Lishu Fault Depression, can be subdivided into southern, central, and northern segments. Synsedimentary slip-rate analysis [23] (Fig 5) reveals pronounced along-strike variability and an overall episodic pattern of fault activity, characterized by a weak-strong-weak evolutionary trend during rifting. During the initial rifting stage, the southern and central segments display comparable slip rates ranging from 75 to 250 m/Ma, whereas the northern segment remains relatively inactive. In the early strong-rifting stage, slip rates in the southern and northern segments decrease to 25–60 m/Ma, while the central segment continues to exhibit high slip rates of 75–250 m/Ma. Fault activity reaches its peak during the late strong-rifting stage, when deformation is dominated by the central segment, which attains maximum slip rates of up to 450 m/Ma. During deposition of the Yingcheng Formation, the southern and northern segments also reach their highest slip rates, approximately 160 m/Ma and 140 m/Ma, respectively. Following the onset of rift attenuation, fault activity diminishes markedly, with slip rates decreasing to 0–60 m/Ma in the southern segment and to 60–300 m/Ma in the central and northern segments. This temporal evolution reflects pronounced fluctuations in fault activity along strike and indicates a progressive northward migration of the basin depocenter.

The Qinjiatun Fault exhibits a complementary pattern of activity (Fig 5b), characterized by peak slip rates of up to 150 m/Ma during the initial rifting stage, a decline in activity during the early strong-rifting stage, renewed faulting in the late strong-rifting stage, and eventual cessation during rift attenuation.

Integrated analysis of the Sangshutai and Qinjiatun faults underscores their fundamental control on the multistage evolution of the Lishu Fault Depression. During the initial rifting stage, isolated fault segments generated discrete clusters of sags, corresponding to deposition of the Huoshiling Formation (Fig 5c). The early strong-rifting stage is characterized by selective fault linkage and progressively increasing fault activity, leading to enhanced structural connectivity (Fig 5d). Continued fault propagation during the late strong-rifting stage resulted in the development of a fully integrated fault system and the generation of maximum accommodation space (Fig 5e). In contrast, during the rift attenuation stage, a reduction in tectonic stress caused fault activity to diminish markedly, and boundary faults ceased to exert a primary control on sediment routing and basin architecture (Fig 5f).

## Paleogeomorphology characteristics

During the initial rifting stage, activation of basin-bounding faults produced a distinctive geomorphic pattern characterized by alternating uplifts and depressions, resulting in a structural framework dominated by multiple grabens. Intense synsedimentary faulting promoted the development of the Qinjiatun, Jinshan, Jingang, and Sangshutai sags along the eastern, southern, and western margins of the basin. The Sangshutai Sag is oriented approximately north-south, with its depocenter located along the central segment of the Sangshutai Fault. In contrast, the Jinshan Sag preferentially developed adjacent to the southern segment of the same fault, whereas the Jingang Sub-sag is comparatively limited in extent and primarily controlled by displacement along the Jingang active fault. The Qinjiatun Sag trends northeast-southwest and is governed by recurrent movement along the Qinjiatun Fault (Fig 6a).

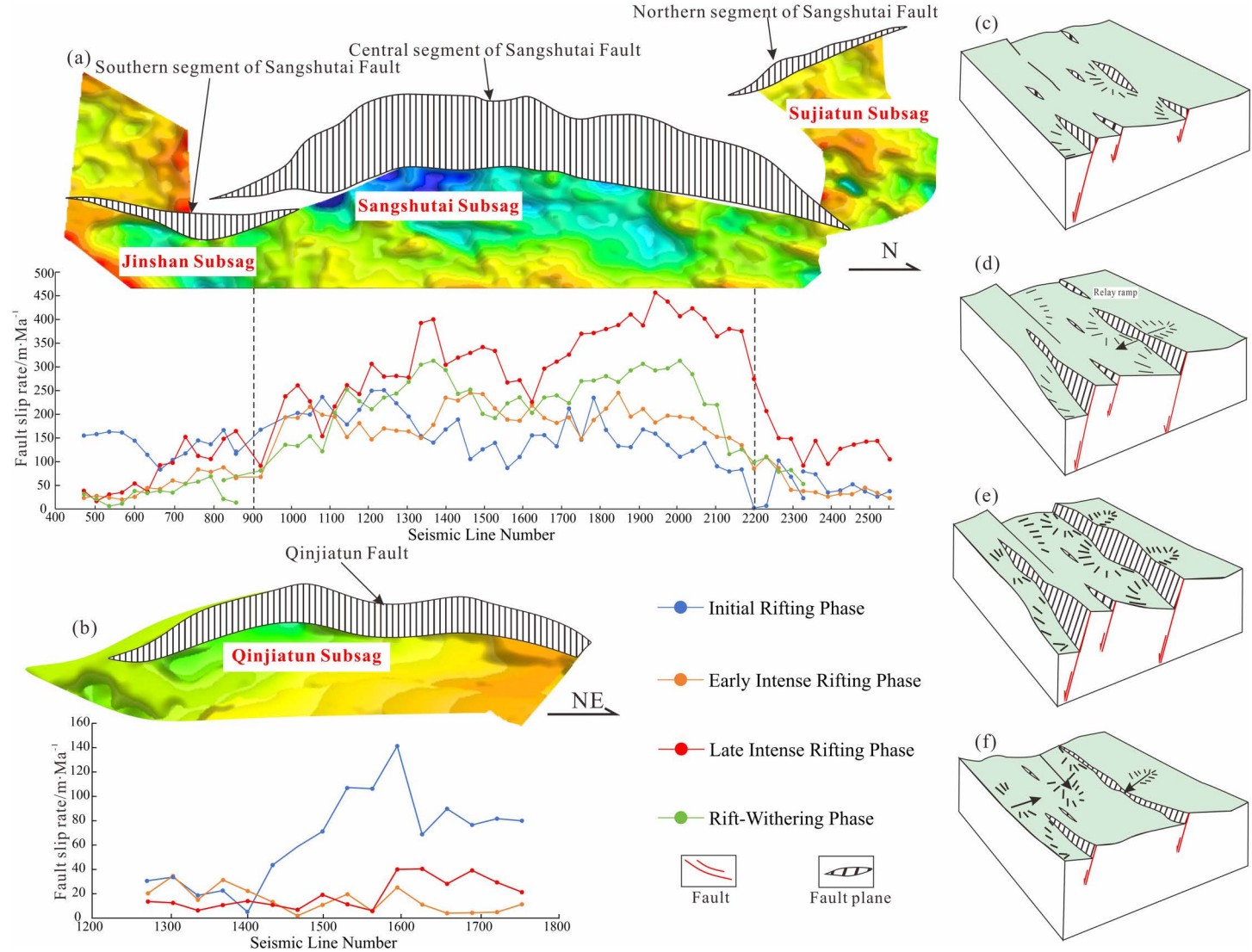

**Fig 5. (a) Statistics of fault activity of the Sangshutai Fault in the Lishu Fault Depression; (b) Statistics of fault activity of the Qinjiatun Fault in the Lishu Fault Depression; (c) Basin evolution model in the initial rifting period; (d) Basin evolution model in the early strong rifting period; (e) Basin evolution model in the late strong rifting period; (f) Basin evolution model in the rifting atrophy period [40].**

During the early stage of intense rifting, both the displacement magnitude and lateral extent of the boundary faults increased markedly. Enhanced fault activity facilitated the linkage of previously isolated sub-sags, ultimately leading to the formation of a unified lacustrine basin. The principal depocenter remained focused along the central segment of the Sangshutai Fault. At the same time, pronounced uplift developed in the southern part of the basin, resulting in the progressive attenuation of the Jingang Sub-sag and the initiation of the Sujiatun Sub-sag (Fig 6b).

During the late stage of intense rifting, the lacustrine basin continued to expand in response to peak activity along the basin-bounding faults. Depocenters migrated progressively northward, and significant basin deepening occurred in areas adjacent to the major boundary faults (Fig 6c).

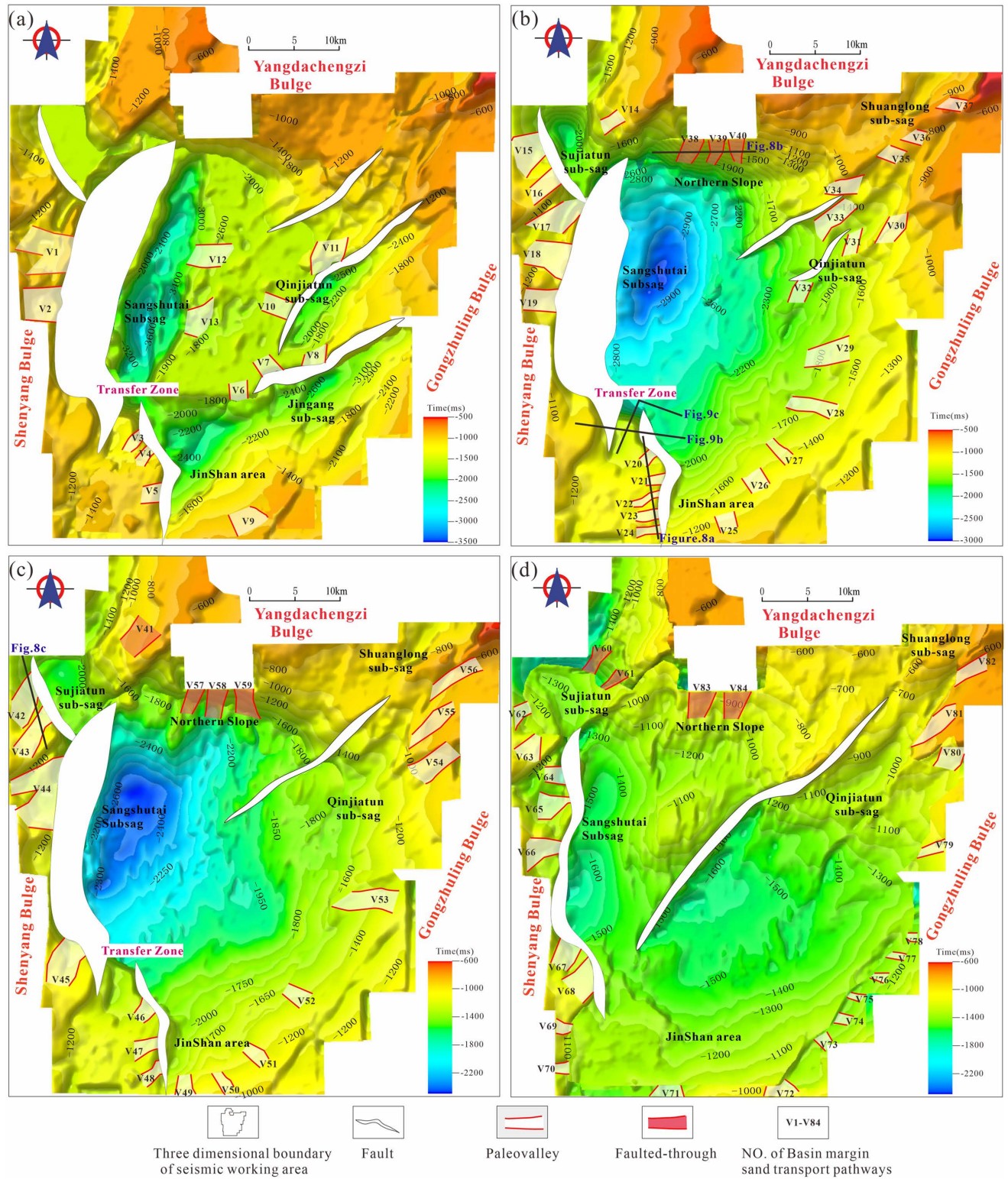

**Fig 6. Paleogeomorphic characteristics and distribution of sediment transport channels of the Jurassic-Cretaceous in the Lishu Fault Depression.** (a) Initial rifting period;(b) Early strong rifting period;(c) Late strong rifting period;(d) Rifting atrophy period. (Note: **Fig 6** was generated using Petrel 2022 and subsequently imported into CorelDRAW for final editing and illustration).

In the waning-rift (rift-attenuation) stage, fault activity declined substantially, and basin extension gradually weakened before ultimately ceasing. Consequently, geomorphic relief within the lake basin was reduced, and the overall depositional area expanded while becoming increasingly subdued (Fig. 6d).

## Source system analysis

From the initial rifting stage through the rift-atrophy stage in the Lishu Fault Depression, seven distinct sediment source areas (A-G) were identified (Table 1): A, the western Yangdachengzi Uplift; B, the northern Shenyang Uplift; C, the southern Shenyang Uplift; D, the southern Jinshan Uplift; E, the eastern Yangdachengzi Uplift; F, the Gongzhuling Uplift; and G, the central uplift of the Lishu Fault Depression.

During the initial rifting stage, four isolated sags developed within the basin. Heavy-mineral assemblages from the Heshan-3 and Heshan-4 wells are dominated by epidote, zircon, and garnet, indicating a primary sediment supply from area F (the Gongzhuling Uplift), which is composed predominantly of metamorphic rocks. In contrast, heavy-mineral assemblages in the Jin-2 well are characterized by zircon, epidote, magnetite, and cassiterite, reflecting provenance from the central uplift of the Lishu Fault Depression (area G), where intermediate-acid magmatic rocks, metamorphic rocks, and tuff constitute the main source lithologies. Sediments from the Li-6 well are dominated by epidote, magnetite, garnet, and zircon, suggesting derivation from the southern Jinshan Uplift (area D), also composed mainly of metamorphic rocks. The Jin-1 well contains assemblages of epidote, magnetite, garnet, zircon, and sphene, indicating sediment input primarily from area C (the southern Shenyang Uplift), with metamorphic and intermediate-acid magmatic rocks as dominant parent materials. Notably, sandstones deposited during this stage are predominantly lithic-rich, reflecting short transport distances and near-source deposition under early rift conditions (Fig 7a).

During the early and late strong-rifting stages, source differentiation became less pronounced, and six sediment source areas (A-F) were identified based on paleogeomorphic reconstruction. In the northern Sujiatun area, heavy-mineral assemblages are dominated by epidote, magnetite, garnet, and zircon, whereas the southern Sujiatun area contains assemblages characterized by epidote, zircon, garnet, and hornblende. These assemblages indicate a mixed provenance system, with sediment contributions derived from both area A (the Yangdachengzi Uplift) and area B (the Shenyang Uplift). Sandstones deposited during this interval are predominantly lithic-feldspathic and feldspathic-lithic in composition. Along the Northern Slope Belt, sandstone compositions are relatively homogeneous, and heavy-mineral assemblages are dominated by zircon, epidote, magnetite, cassiterite, sphene, and garnet, indicating derivation primarily from the northern Yangdachengzi Uplift, which consists of mixed metamorphic and magmatic source rocks. In the Qinjiatun area, heavy-mineral assemblages composed of epidote, magnetite, zircon, garnet, and sphene suggest sediment supply from area F (the Gongzhuling Uplift), characterized by acid magmatic and volcanic rocks; sandstones in this area are mainly lithic-feldspathic. The Jinshan area receives sediment input from both western and southern source regions, as indicated by heavy-mineral assemblages including epidote, magnetite, garnet, zircon, sphene, and cassiterite. Sandstone compositions in this area exhibit a progressive transition from lithic-rich to feldspathic-lithic types, reflecting increasing transport distances and enhanced sediment reworking during strong rifting (Figs 7b-c).

**Table 1. Heavy mineral assemblage characteristics and potential source areas in the Lishu Fault Depression.**

| No. | Heavy Mineral Assemblage | Parent Rock Lithology | Potential Source Area |
|---|---|---|---|
| 1 | Epidote, Zircon, Garnet, Magnetite, Sphene | Acidic Igneous Rock, Volcanic Rock | F(Gongzhuling Uplift)、C(South of Shenyang Uplift) |
| 2 | Zircon, Epidote, Magnetite, Cassiterite | Int.-Acid. Igneous, Schist, Tuff | G(Central Uplift of Lishu Depression) |
| 3 | Epidote, Mag., Garnet, Zircon, Sphene, Cas | Int.-Acid. Igneous, Metamorphic | D(South Jinshan Uplift) |
| 4 | Epidote, Magnetite, Garnet, Zircon | Metamorphic, Igneous Rock | A、E(Yangdachengzi Uplift) |
| 5 | Epidote, Zircon, Garnet, Hornblende | Int.-Acid. Igneous,Metamorphic | B、C(Shenyang Uplift) |

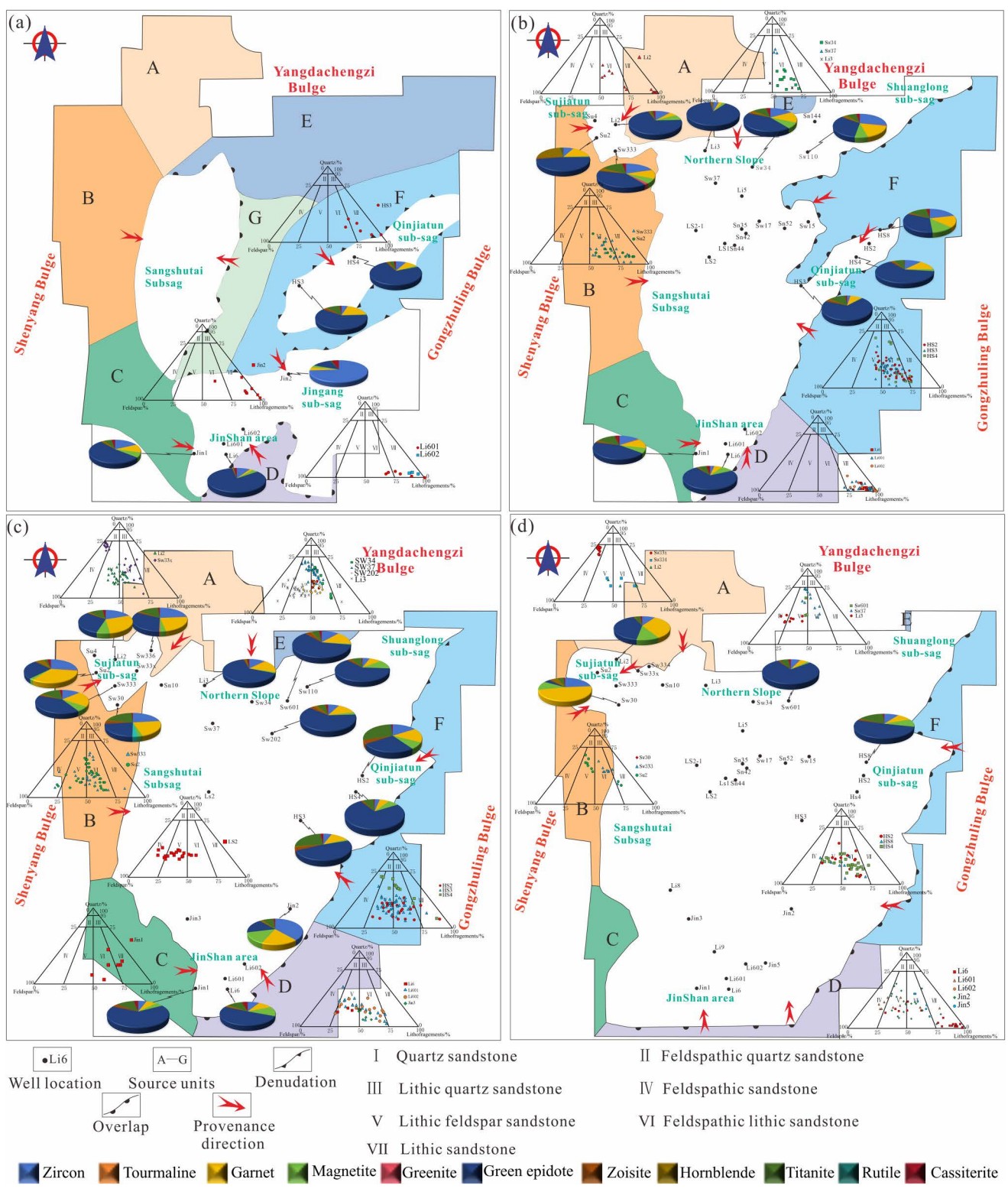

**Fig 7. Petrological characteristics and source division units of the Jurassic-Cretaceous in the Lishu Fault Depression.** (a) Initial rifting period; (b) Early strong rifting period; (c) Late strong rifting period; (d) Rifting atrophy period. (Note: Fig 7 was generated using Petrel 2022 and subsequently imported into CorelDRAW for final editing and illustration).

During the rifting-atrophy stage, the spatial extent of sediment source areas contracted slightly; however, the dominant sediment supply directions remained largely unchanged. In the northern Sujiatun area, heavy-mineral assemblages continue to be dominated by epidote, magnetite, garnet, and zircon, whereas the southern Sujiatun area is characterized by assemblages of epidote, zircon, garnet, and hornblende, confirming the persistence of a mixed-source provenance system. Sediment input in this region is derived primarily from area A (the Yangdachengzi Uplift) and area B (the Shenyang Uplift). Along the Northern Slope Belt, sediment supply continues to be dominated by the northern Yangdachengzi Uplift, composed of mixed metamorphic and magmatic source rocks. Sandstones in this area are predominantly lithic-feldspathic in composition. The Qinjiatun area remains sourced mainly from area F (the Gongzhuling Uplift), characterized by acid magmatic and volcanic rocks, and associated sandstones retain a lithic-feldspathic composition. In the Jinshan area, sediment input from western and southern source regions persists, as indicated by heavy-mineral assemblages dominated by epidote, magnetite, garnet, zircon, sphene, and cassiterite. Sandstones exhibit an increased feldspar content relative to earlier stages, reflecting longer transport distances and more distal-source characteristics during rift attenuation (Fig 7d).

## Transport systems analysis

**Development characteristics of valleys.** The margins of the Lishu Fault Depression have undergone prolonged exposure, differential weathering, and denudation, resulting in the development of drainage systems characterized by distinct channel types. High-resolution seismic data were employed to delineate the spatial distribution, geometry, and scale of paleo-valleys and to identify syndepositional faults (Fig 8). In total, 84 valley systems, including both sediment-incised valleys and fault-trough-controlled valleys, developed along the Jurassic-Cretaceous basin margins.

During deposition of the Shahezi Formation, U-shaped, V-shaped, and W-shaped valleys developed along the basin margins, exhibiting clear spatial regularities in their distribution. Valleys formed during deposition of the first member of the Shahezi Formation (V20-V24), for example, are concentrated along the southern segment of the Sangshutai Fault within a steep-slope setting. In this area, U-, V-, and W-shaped morphologies predominate. On seismic profiles, V-shaped valleys are typically expressed as isolated, short-axis or discontinuous reflection packages. Downslope, within areas of relatively lower potential energy, weakened paleohydrodynamic conditions commonly promote a transition from narrow V-shaped

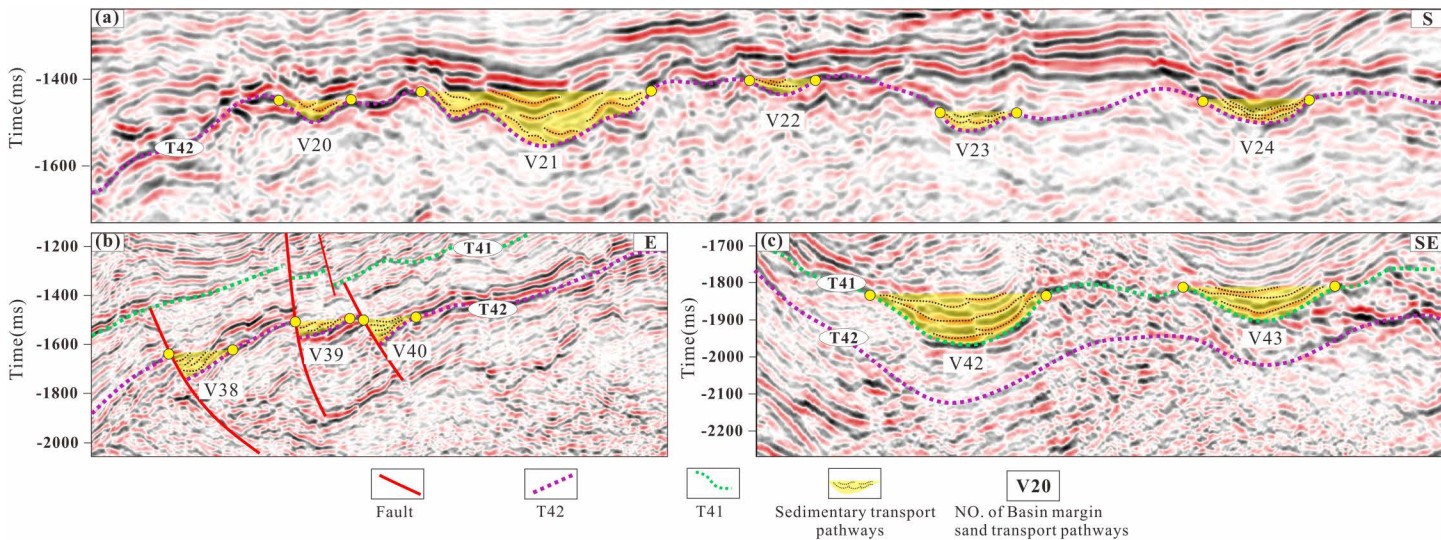

**Fig 8. Seismic reflection characteristics of the basin-margin valleys in the Lishu Fault Depression (The location of the seismic profile is shown in Figs 6b and c).**

valleys to broader U-shaped forms. This morphological change is accompanied by increases in valley width and depth, as well as vertical stacking and lateral migration of seismic reflection axes.

Toward the lacustrine depocenter, the main valley axes become increasingly influenced by water-body support, resulting in reduced channel confinement and weakened hydrodynamic energy. Under these conditions, lateral tributaries exhibit diminished activity, seismic isochrons become more continuous, and W-shaped paleo-valleys become the dominant valley type, as exemplified by valley V21 (Fig 8). In contrast, valleys V38-V39 along the northern slope represent fault-trough-type systems generated by active syndepositional faulting, which created localized negative relief favorable for sediment routing and accumulation. Valleys V42-V43 in the Sujiatun area within the Yingcheng Formation are characterized by broad, gently incised U-shaped morphologies and highly continuous seismic isochrons.

Overall, early-stage valley development within high-relief, steep-slope belts is dominated by incision and vertical down-cutting, commonly producing narrow V-shaped valleys characterized by strong hydrodynamic energy and limited tributary connectivity. Downstream, as hydrodynamic energy progressively decreases, broader U-shaped valleys become dominant, reflecting efficient sediment transport and enhanced sand delivery capacity. During the late evolutionary stages of the paleo-drainage system, flow bifurcation combined with further reductions in hydrodynamic energy promotes the development of W-shaped valleys. These valleys exhibit reduced sediment transport efficiency and are typically associated with proximal sediment unloading and accumulation near the basin margin (Fig 8).

**Development characteristics of transfer zones.** A transfer zone is a structural domain that accommodates regional extensional strain by transferring fault displacement between adjacent fault segments [40] (Fig 9a). Such zones commonly manifest as changes in fault strike or as overlapping fault segments near fault tips (Gawthorpe and Leeder, 2000). In the study area, the transfer zone develops where the southern and central segments of the Sangshutai Fault overlap laterally, coinciding with a progressive reduction in displacement toward the segment tips (Fig 9a).

At the dip terminations of these fault segments, a transfer ramp develops to accommodate along-strike variations in fault throw (Fig 9d). This ramp connects the hanging wall of the southern segment with the footwall of the central segment, and its strike is approximately perpendicular to that of the basin-bounding fault [41,42]. In vertical dip-oriented seismic sections, the two fault segments appear subparallel and bound a zone of reduced stratigraphic thickness (Fig 9b), whereas fault displacement increases progressively toward the interiors of both segments. Along the ramp profile, prominent progradational seismic reflection patterns are observed, indicating enhanced sediment delivery and depositional infill across the transfer zone (Fig 9c).

**Parallel fault-step zones.** A parallel fault-step zone is a structural configuration characterized by multiple faults of comparable scale and generally consistent strike arranged in a subparallel pattern. Such configurations typically develop under regional extensional regimes that generate a series of step-like, parallel fault strands. In the eastern part of the Lishu Fault Depression, particularly within the Qinjiatun Sub-depression, this structural style is well developed. From the basin margin toward the basin interior, a succession of fault-controlled terraces forms, producing a gently dipping fault-step geometry in cross section. During the rifting stage, differential subsidence among fault blocks within the parallel fault system generated a distinctive multi-step paleogeomorphic surface. This structural evolution governed spatiotemporal variations in accommodation space and, consequently, exerted a strong control on the location of sedimentary systems and the spatial distribution of sand bodies (Fig 10).

## Depositional system analysis

### Main sedimentary facies types.

(1) Near-shore Subaqueous Fan

Nearshore subaqueous fans represent gravity-flow-dominated depositional systems that develop along the steep slope belts of lacustrine basins adjacent to sediment source areas in the Lishu Fault Depression. These systems are composed

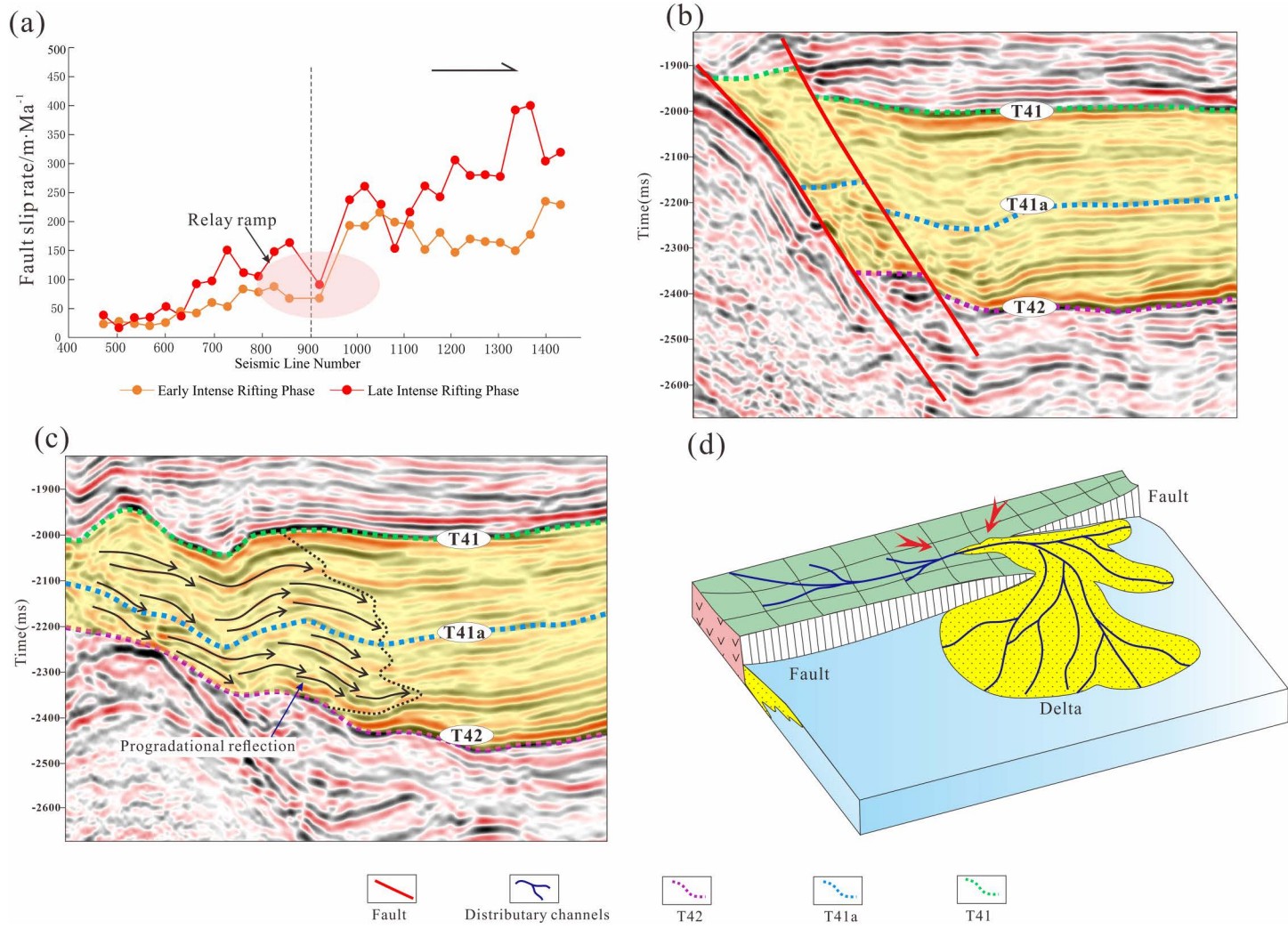

**Fig 9. (a) Statistics of fault activities in the early and late strong rifting periods of the southern and middle sections of the Sangshutai Fault in the Lishu Fault Depression.** (b) Vertical profile of the southern and middle sections of the Sangshutai Fault in the Lishu Fault Depression;(c) Profile of the basin entry along the southern and middle sections of the Sangshutai Fault in the Lishu Fault Depression;(d) Transfer zone model of the southern and middle sections of the Sangshutai Fault in the Lishu Fault Depression (The location of the seismic profile is shown in Fig 6b).

predominantly of gravel-rich deposits generated by gravity-flow processes and commonly display diagnostic sedimentary structures, including graded bedding, massive bedding, and slump deformation. In plain view, nearshore subaqueous fans exhibit a fan-shaped geometry and can be subdivided into fan-root, fan-middle, and fan-end facies belts. For example, in the Shahezi Formation penetrated by the Jin-1 well near the Sangshutai Fault Zone, the dominant lithology is conglomerate, and well-log responses are characterized by thick, box-shaped curves, indicative of nearshore subaqueous fan braided-channel deposits. Vertically, these deposits display stacked "thick sandbody" cycles formed by the superposition of multiple fan-building events. Because fan sediments are rapidly transported into deep-lacustrine settings, carbonaceous laminae are locally preserved, and convolute bedding and conglomeratic sandstone are commonly developed. These features reflect near-source, deep-water rapid accumulation under high sedimentation rates and strong depositional heterogeneity, commonly resulting in reservoir sand bodies with pronounced pore-permeability contrasts (Figs 11a, b). In

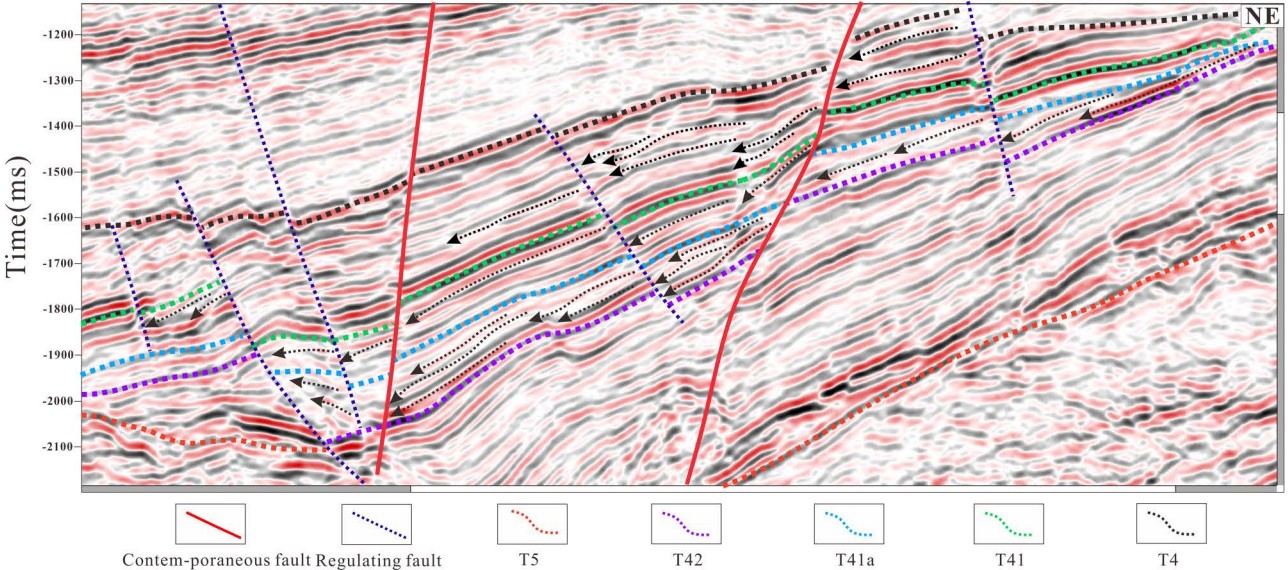

**Fig 10. Profile of parallel faults on the eastern gentle slope of the Lishu Fault Depression (The location of the seismic profile is shown in Fig 1a).**

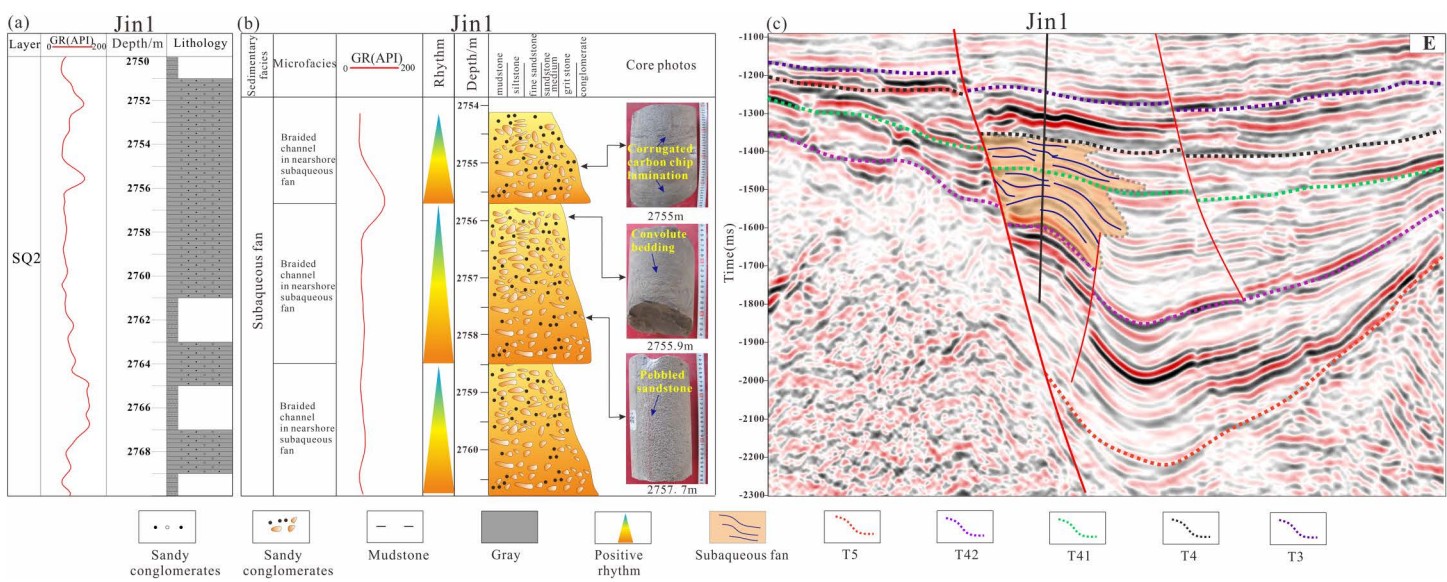

**Fig 11. (a) Log facies histogram of the Shahezi Formation in Well Jin1 of the Lishu Fault Depression; (b) Core facies histogram of the Shahezi Formation in Well Jin1 of the Lishu Fault Depression;(c) Seismic facies characteristics across Well Jin1 in the Lishu Fault Depression.** (The location of the seismic profile is shown in Fig 1a).

seismic profiles, nearshore subaqueous fans typically appear as wedge-shaped depositional bodies accumulated at the base of steep basin slopes, exhibiting an overall divergent reflection geometry in which reflections converge toward the fan terminus. In some cases, deeper portions of the fan bodies are modified by fault activity and display chaotic, mound-shaped seismic reflections (Fig 11c).

## (2) Fan Delta

The fan-delta plain subfacies represents the terrestrial component of the fan-delta depositional system and typically develops within the proximal foreland transitional zone. This subfacies is dominated by coarse-grained clastic deposits. In the Jinshan area, for example, sediments penetrated by Well Li-6 consist primarily of gravelly sandstone and conglomerate, bounded by a well-defined erosional surface. The associated reddish-brown mudstones display clear oxidized coloration, and wireline logs are characterized by a box-shaped motif, indicative of channelized deposition. Fluvial-channel and interchannel subfacies are well developed (Figs 12a, b). Vertically, the succession exhibits a normal-grading depositional cycle, with sharp basal contacts against underlying oxidized mudstones. Seismically, the fan-delta plain is characterized by medium- to weak-amplitude reflections; locally, reflection-poor or blank zones occur, likely reflecting coarse lithologies and strong heterogeneity within the deposits (Fig 12c).

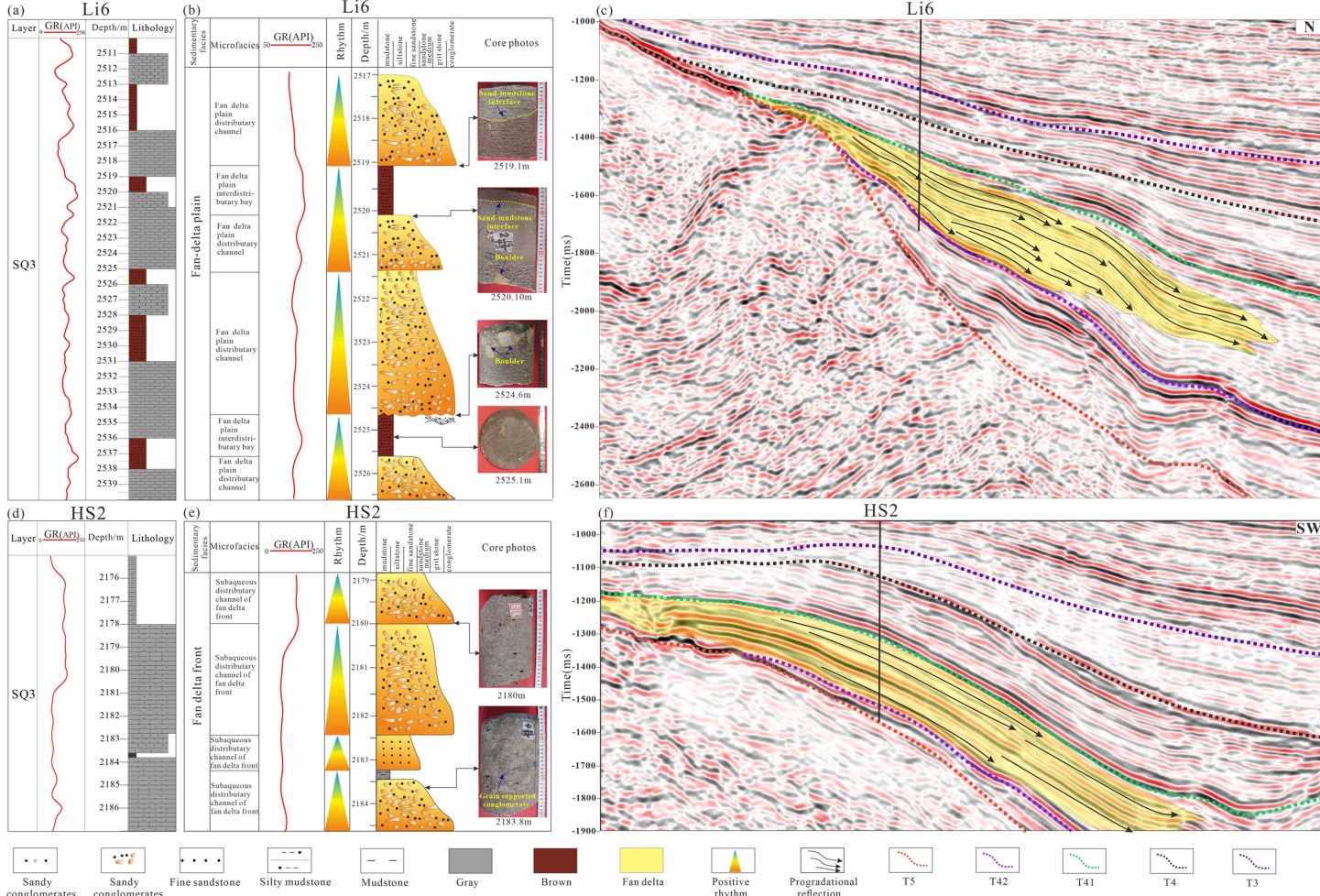

**Fig 12.** **(a) Log facies histogram of the Shahezi Formation in Well Li6 of the Lishu Fault Depression; (b) Core facies histogram of the Shahezi Formation in Well Li6 of the Lishu Fault Depression; (c) Seismic facies characteristics across Well Li6 in the Lishu Fault Depression (The location of the seismic profile is shown in Fig 1a); (d) Log facies histogram of the Shahezi Formation in Well HS2 of the Lishu Fault Depression; (e) Core facies histogram of the Shahezi Formation in Well HS2 of the Lishu Fault Depression; (f) Seismic facies characteristics across Well HS2 in the Lishu Fault Depression (The location of the seismic profile is shown in Fig 1a).**

The fan-delta front subfacies occupies the transitional zone between the fan-delta plain and deeper lacustrine environments. In the Qinjiatun area, data from Well HS2 indicate that the second sand member (S2) is composed predominantly of fine sandstone to gravelly sandstone, interbedded with thin gray-black mudstone layers. Locally, conglomerates exhibit particle-supported textures (Figs 12d, e), and wireline logs display a thick, box-shaped response characteristic of channelized deposition. Vertically, the succession records multiple stages of subaqueous distributary-channel infill. On the northern slope, the S2 interval in Well Li3 is dominated by silty to fine-grained sandstone and gray-black mudstone, and includes microfacies such as sheet sands and distributary-bay deposits (Fig 12f). Common sedimentary structures include parallel lamination and carbonaceous laminae. Seismically, the fan-delta front is characterized by steeply dipping progradational reflection configurations with medium- to strong-amplitude responses, collectively indicating a high-energy depositional environment.

(3) Braided River Delta

Braided River Delta Plain. This subenvironment is characterized by multiple braided distributary channels with well-developed basal scour surfaces, mud-gravel lag deposits, and large-scale trough and planar cross-stratification. The deposits are dominated by medium- to coarse-grained sandstones and commonly exhibit a complete upward-fining succession, locally preceded by basal coarsening related to channel incision and reoccupation (Fig 13a). Overall, the vertical stacking pattern reflects repeated channel migration and aggradation. Wireline log responses are typified by low gamma-ray (GR) values with box-shaped to bell-shaped motifs, indicating stacked channel-fill sand bodies with good reservoir potential (Fig 13b). Seismically, this subfacies displays moderate- to weak-amplitude reflections with discontinuous internal architectures and locally divergent to subparallel reflection configurations, consistent with laterally migrating braided-channel complexes (Fig 13c).

Braided River Delta Front. This subfacies is characterized by subaqueous distributary channels developed at the delta front, where sandstone grain size typically decreases to medium-fine and sediment color becomes darker, reflecting increased water depth and reduced oxidation. Large-scale inclined bedding and parallel lamination are common, and lateral channel migration generates wedge-shaped or sigmoidal cross-stratification. Tidal-bar deposits are generally absent, whereas mudstone content increases markedly within interdistributary bay environments, indicating reduced hydrodynamic energy and enhanced suspension settling. GR log responses exhibit diverse patterns, including bell-shaped, box-shaped, funnel-shaped, and finger-like motifs, with pronounced amplitude variability. Subaqueous distributary-channel fills are commonly expressed as high-amplitude box- or bell-shaped log signatures, reflecting relatively clean sand bodies, whereas interdistributary bays are characterized by low-amplitude, serrated ("tooth-shaped"), or near-baseline responses indicative of fine-grained sediment accumulation (Figs 13d and e). Seismic facies display diagnostic oblique-accretion geometries, commonly expressed as shingle-like or broom-shaped reflection patterns, with moderate to strong amplitudes. Reflections are continuous to semi-continuous and typically exhibit onlapping configurations with wedge-shaped or lens-shaped external morphologies. Vertically sourced sediment dispersal and widespread sheet-flow deposition produce laterally extensive, sheet-like reflection packages with predominantly subparallel internal architectures (Fig 13f).

(4) Lake Facies

Shore-shallow lake deposits are dominated by fine-grained sediments, primarily comprising dark mudstone, siltstone, and calcareous shale. These successions commonly exhibit horizontal bedding, thin lamination, and pervasive bioturbation structures, indicating deposition under relatively low-energy, shallow lacustrine conditions. The sediments are generally well sorted and texturally mature, reflecting prolonged reworking and limited clastic influx. In Well SN48, core observations reveal abundant plant stem fragments, suggesting intermittent terrestrial influence and nearshore vegetation input. Correspondingly, wireline logs display low-amplitude, serrated to near-baseline responses, consistent with fine-grained, organic-rich deposits (Figs 14a and b). Seismically, these deposits are characterized by medium-amplitude,

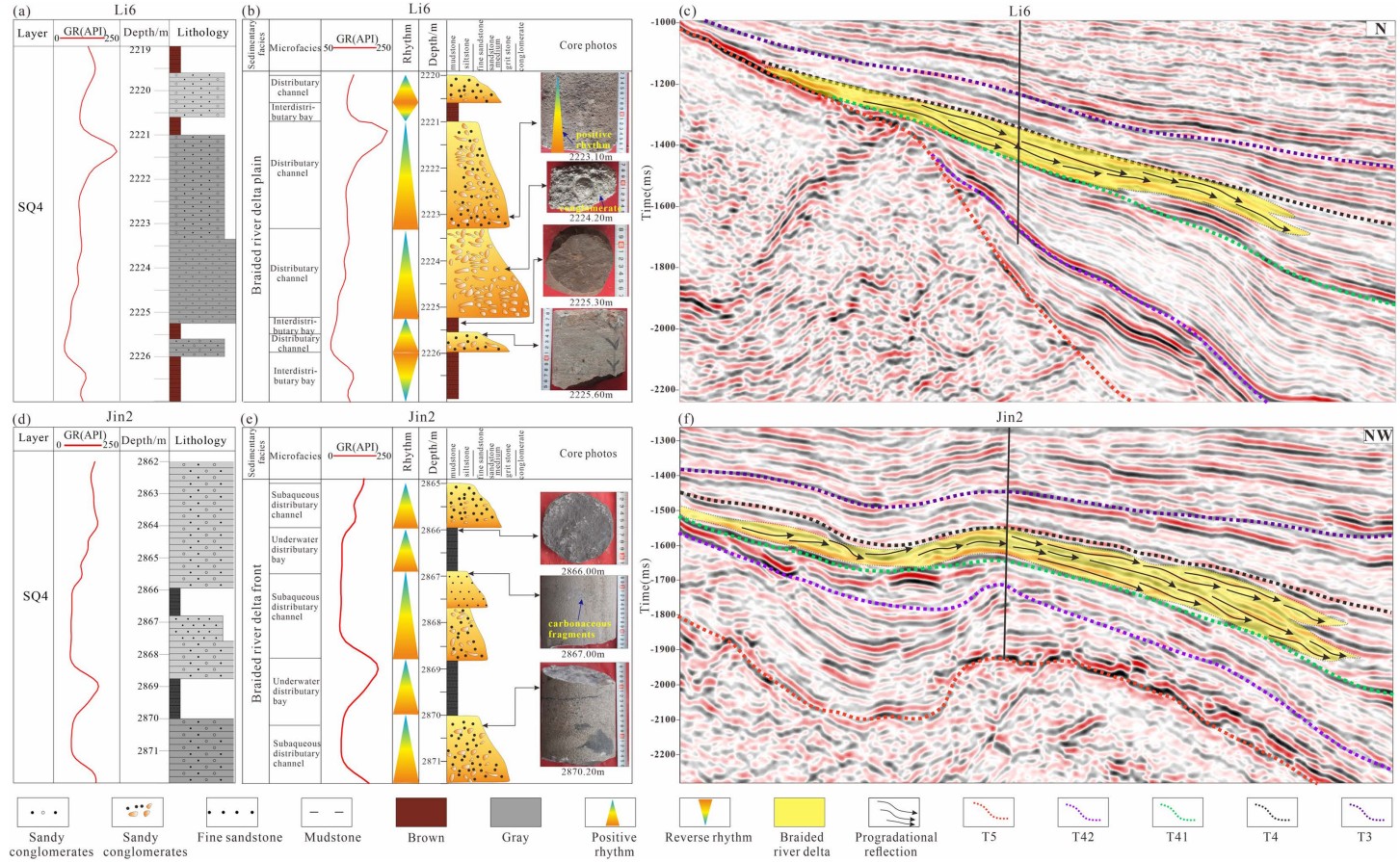

**Fig 13.** (a) Log facies histogram of the Yingcheng Formation in Well Li6 of the Lishu Fault Depression;(b) Core facies histogram of the Yingcheng Formation in Well Li6 of the Lishu Fault Depression;(c) Seismic facies characteristics across Well Li6 in the Lishu Fault Depression (The location of the seismic profile is shown in Fig 1a);(d) Log facies histogram of the Yingcheng Formation in Well Jin2 of the Lishu Fault Depression;(e) Core facies histogram of the Yingcheng Formation in Well Jin2 of the Lishu Fault Depression;(f) Seismic facies characteristics across Well Jin2 in the Lishu Fault Depression (The location of the seismic profile is shown in Fig 1a).

medium-frequency, subparallel reflections with good lateral continuity, reflecting laterally extensive, relatively uniform sedimentation within the shore-shallow lake environment (Fig 14e).

Deep-lake facies develop below the storm wave base and represent the deepest depositional environments within the lacustrine basin. These deposits are dominated by gray-black to dark-gray mudstone and shale, characterized by very fine grain size and relatively high organic-matter content, reflecting low-energy conditions and sustained suspension settling. In the second member of the Shahezi Formation in Well LS2, located within the Sangshutai deep sag, the succession is composed primarily of gray-black mudstone interbedded with muddy siltstone. Wireline logs exhibit a notably smooth, low-variability response, consistent with fine-grained, homogeneous lithologies (Figs 14c and d). Seismically, the deep-lake deposits are expressed by strong-amplitude, medium- to low-frequency, laterally continuous parallel reflections, indicative of widespread, uniform sediment accumulation under stable deep-lacustrine conditions (Fig 14e).

**Spatiotemporal distribution characteristics of depositional systems.** During the initial rifting stage, fan-delta systems interbedded with volcanic clastic deposits developed along the margins of individual, fault-controlled lacustrine depocenters. Volcanic rocks commonly occur at the base of these successions, particularly in uplifted areas, reflecting

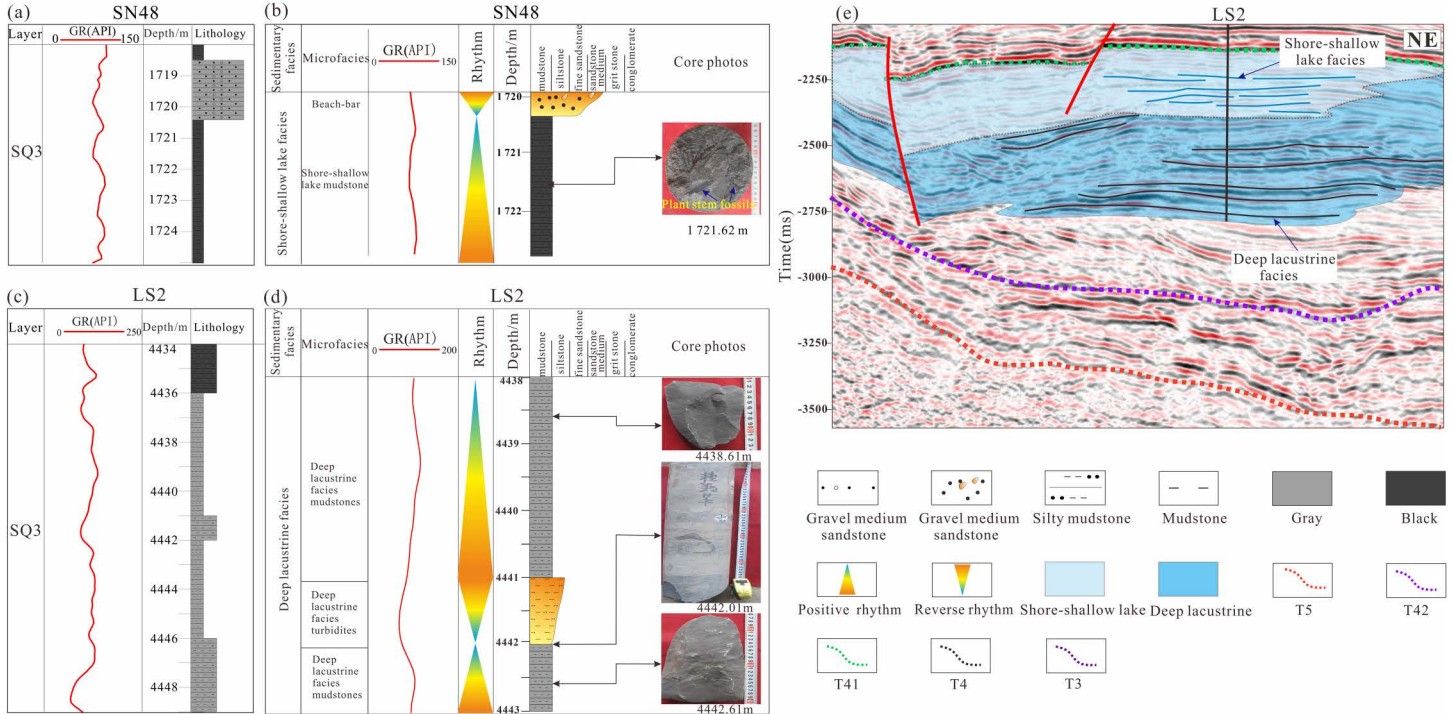

**Fig 14. (a)** Log facies histogram of the Shahezi Formation in Well SN48 of the Lishu Fault Depression;**(b)** Core facies histogram of the Shahezi Formation in Well SN48 of the Lishu Fault Depression;**(c)** Log facies histogram of the Shahezi Formation in Well LS2 of the Lishu Fault Depression;**(d)** Core facies histogram of the Shahezi Formation in Well LS2 of the Lishu Fault Depression;**(e)** Seismic facies characteristics across Well LS2 in the Lishu Fault Depression (The location of the seismic profile is shown in **Fig 1a**).

contemporaneous volcanic activity associated with early extensional tectonism. Meanwhile, several localized fault depressions, such as the Jinshan, Jingang, Qinjiatun, Shuanglong, and Sujiatun sags, began to receive clastic input, forming stratigraphic successions characterized by interlayered volcanic and siliciclastic deposits. Sand bodies sourced from the eastern sub-sag are clearly identifiable in cross-sectional profiles, indicating active sediment supply from adjacent uplifted source areas. During this stage, the lacustrine basin was relatively small and shallow. Most depocenters were dominated by shore-shallow lake facies, whereas deeper-water conditions were restricted to the western part of the basin, west of the basin-controlling Sangshutai Fault. In this area, a nearshore subaqueous fan system developed, reflecting localized deepening and enhanced accommodation adjacent to the active boundary fault (Figs 15 and 16a).

During the intensified rifting stage (Shahezi-Yingcheng), lake levels rose markedly during deposition of the Shahezi Formation. Extensive semi-deep to deep-lacustrine facies developed within the fault depressions, recording the maximum lacustrine transgression and constituting the principal interval of source-rock accumulation. These deeper-water deposits expanded outward into the eastern, southern, and northern shallow-lake margins. Along the western basin margin adjacent to the Sangshutai Fault, sedimentation was dominated by nearshore subaqueous fan systems, whereas well-developed fan-delta systems formed along the northern and southeastern slope belts. The Shahezi mudstones are predominantly black, laterally extensive, and relatively thick, reflecting sustained deep-water, low-energy depositional conditions (Figs 15 and 16b). The Yingcheng period represents the structural shaping stage of the half-graben system and largely inherited the depositional framework established during the Shahezi interval. Fan bodies expanded significantly in areal extent, while the overall lake area contracted. Braided-river deltas, nearshore subaqueous fans, and lacustrine facies coexisted during this stage; however, water depths were generally shallower, and mudstones are dominantly gray,

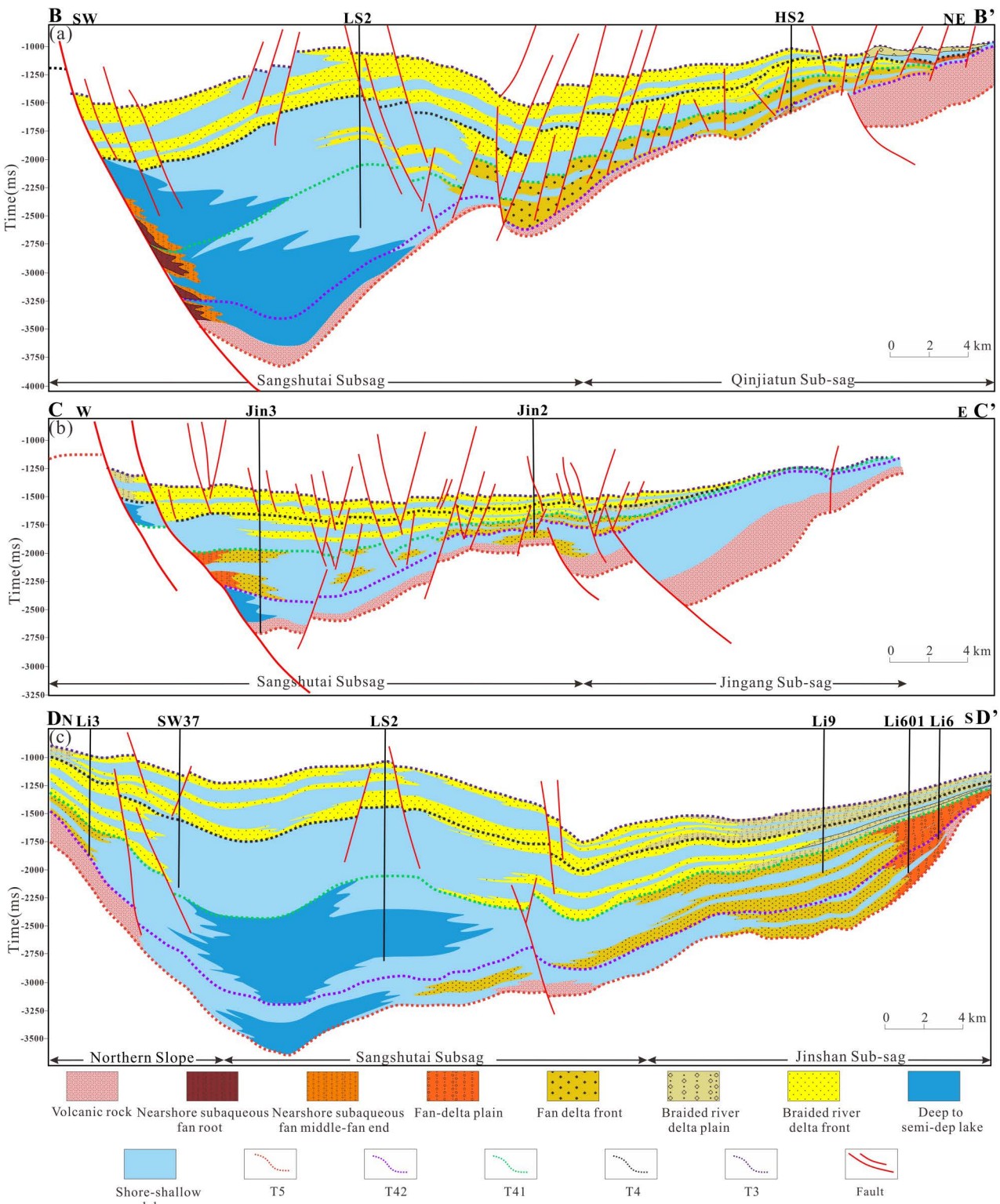

**Fig 15.** (a) Sedimentary filling profile across Wells LS2-HS2 in the Lishu Fault Depression; (b) Sedimentary filling profile across Wells Jin2-Jin3 in the Lishu Fault Depression; (c) Sedimentary filling profile across Wells Li3-SW37-LS2-Li9-Li601-Li6 in the Lishu Fault Depression (The location of the seismic profile is shown in Fig 1a).

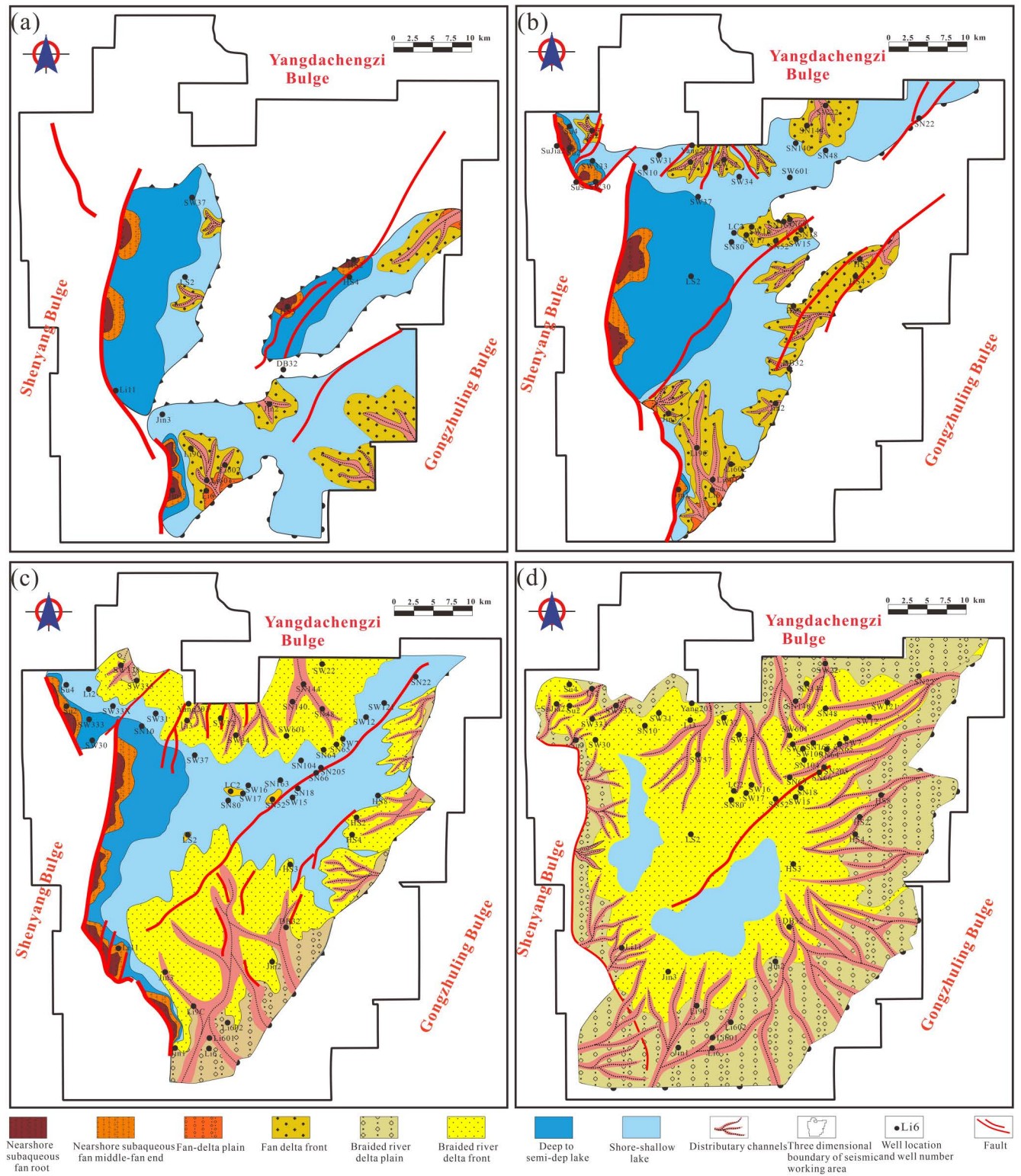

**Fig 16. Distribution of sedimentary systems of the Jurassic-Cretaceous in the Lishu Fault Depression.** (a) Initial rifting period; (b) Early strong rifting period; (c) Late strong rifting period; (d) Rifting atrophy period. (Note: Fig 16 was generated using Petrel 2022 and subsequently imported into CorelDRAW for final editing and illustration).

indicating increased clastic input and enhanced hydrodynamic conditions relative to the preceding Shahezi period (Figs 15 and 16c).

During the waning rift stage, represented by deposition of the Denglouku Formation, tectonic subsidence diminished markedly and the lacustrine basin expanded under an overall shallow-water regime. The depositional system evolved into a broad, laterally extensive shallow-lake to delta complex. Braided-river deltas increased substantially in scale, basin-floor relief became progressively gentler, and east-west-trending fan bodies coalesced in cross-sectional view. Lacustrine facies were preserved only locally within the central part of the fault depression, reflecting reduced accommodation and widespread sediment infilling during rift attenuation (Figs. 15 and 16d).

## Discussion

This study directly addresses the key questions outlined in the Introduction concerning the evolution of the Lishu Fault Depression. Our results demonstrate that episodic rifting exerts a first-order control on sediment accumulation patterns. Variations in fault activity, drainage reorganization, and accommodation development during multiphase rifting fundamentally regulate the geometry, scale, and connectivity of depositional systems. In particular, the distribution and development of nearshore subaqueous fans and deltaic deposits show strong, systematic correlations with valley width, valley depth, and upstream drainage area, underscoring the role of intermittent tectonic pulses in controlling sediment routing efficiency, transport capacity, and basin-scale sediment dispersal.

Second, we develop a quantitative S2S coupling model through the integrated analysis of seismic, well-log, and core data, enabling systematic reconstruction of sediment provenance, transport pathways, and depositional systems. The model explicitly resolves the temporal and spatial evolution of the S2S system by linking variations in fault activity rates to changes in sediment-routing efficiency and depositional architecture. By directly coupling multiphase rifting dynamics with source, transport, and sink components, this framework provides a robust predictive tool for assessing sand-body distribution and for guiding hydrocarbon exploration and reservoir targeting in the Lishu Fault Depression.

### Vertical evolution of source-sink system and sand body development model

Comparison of the Development Models of Typical Rift Basins Worldwide and the Multi-phase Rift Development Model in the Lishu Fault Depression of the Songliao Basin (Fig 17):

During the initial rifting stage, NW-SE-oriented extensional stress produced a series of NNE-trending normal faults, resulting in a domino-style array of small, NE-striking rift basins. Sediment routing during this phase was largely inherited from pre-existing drainage networks, which were locally modified by fault scarps and growth-fold-related paleotopography. Sediment provenance was dominated by low-relief intrabasinal uplifts surrounding the depression, and sediment was transported predominantly through valley-confined routing systems. Along the margins of these nascent rift basins, fan-delta systems composed mainly of volcanic-derived clastic material were preferentially developed, whereas basin interiors were characterized by small, shallow lacustrine environments. In the western steep-slope belt, the water body beneath the Sangshutai basin-controlling fault is relatively deep, leading to the development of a nearshore subaqueous fan depositional system. In the eastern gentle-slope belt and the southern sub-sag, isolated fan-delta depositional systems are dominant. Overall, a differentiated source–to-sink coupling pattern characterized by steep-slope nearshore subaqueous fans and fan-delta systems is developed (Fig 17a).

During the early stage of intensified fault activity, boundary faults continued to propagate under near-east-west extensional stress, and tectonic subsidence became the dominant mechanism generating accommodation space. Sediment supply was derived primarily from the Western Shenyang Uplift, the Northern Yangdachengzi Uplift, and the Eastern Gongzhuling Uplift. At this stage, the Sangshutai Fault of western steep-slope belt evolved into an effective structural transfer zone that redistributed regional stress and strain, the center of subsidence began to migrate northward, thereby exerting a first-order control on sediment-routing pathways and depositional patterns. This tectonic reorganization is

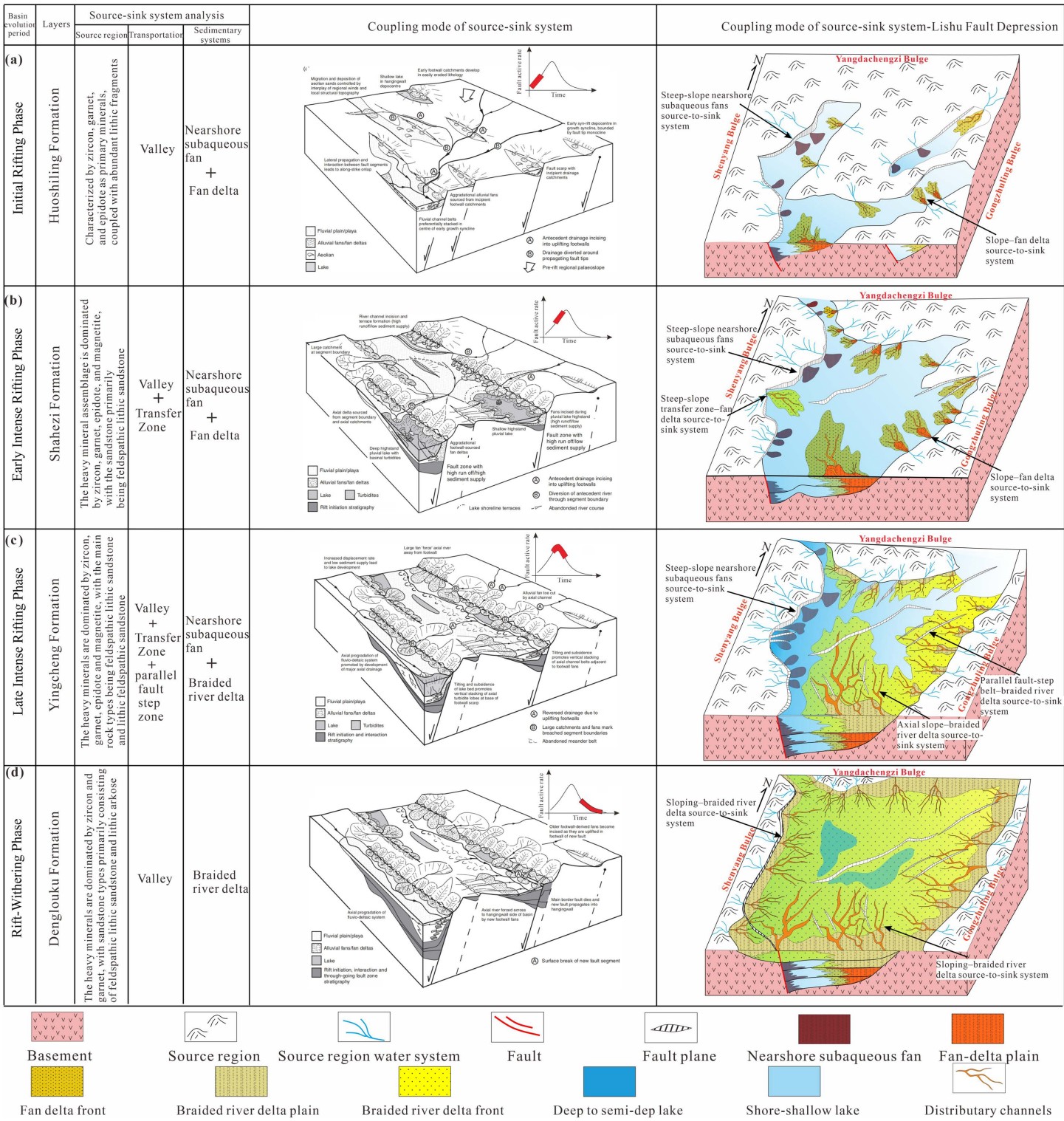

**Fig 17. Coupling model of source-to-sink systems of the Jurassic-Cretaceous in the Lishu Fault Depression.** (a) Initial rifting period; (b) Early strong rifting period; (c) Late strong rifting period; (d) Rifting atrophy period (according to [40], with modifications). (Note: Fig 17 was created using CorelDRAW).

reflected in the modification of the principal fluvial network and a systematic reorientation of sediment dispersal directions. Subaqueous distributary-channel sand bodies preferentially developed along the margins of the transfer zone, recording enhanced sediment throughput. Along the eastern margin of the basin, a series of subparallel faults controlled the development of extensive semi-deep to deep lacustrine facies. During this stage, a pronounced convergence of sediment supply was observed. In the early stage, dominant source–sink coupling systems developed as steep-slope transfer zone–fan delta and axial slope–fan delta configurations, whereas in the eastern gentle slope domain, a parallel fault-step belt–source–sink coupling system was established (Fig 17b).

During the late stage of strong rifting, enhanced interaction and linkage among fault segments promoted significant along-strike propagation of fault arrays. This process led to the expansion and partial coalescence of previously isolated depocenters, while several subsidiary fault segments remained relatively inactive. Sediment provenance patterns remained broadly stable, with the Western Shenyang Uplift, Northern Yangdachengzi Uplift, and Eastern Gongzhuling Uplift continuing to supply the majority of clastic material. In contrast, sedimentation in the southern Jinshan area was increasingly influenced by axial and more distal source regions, where a braided-river-delta system prograded across the central depression. Within the central segment of the Sangshutai Fault, the depocenter continued to migrate northward, this evolutionary stage culminated in the establishment of a strongly asymmetric rift architecture, characterized by a dominant western boundary fault and progressive eastward basin expansion. Under these conditions, nearshore subaqueous fans and braided-river delta deposits became the principal sedimentary systems. The lacustrine basin remained relatively shallow, and mudstone deposits are predominantly gray, reflecting reduced water depth and enhanced clastic input. In this stage, a transverse gentle-slope setting developed a parallel step-fault zone–braided river delta source-to-sink system, while the axial fluvial system was dominant, giving rise primarily to an axial slope–braided river delta source-to-sink system (Fig 17c).

During the subsequent rift-contraction stage, previously isolated fault segments became fully linked, establishing a through-going fault system that functioned as a major structural transfer zone. Displacement along these connecting faults reconfigured intrabasinal uplifts and removed former topographic barriers, allowing axial fluvial systems to traverse areas that had previously separated individual depocenters. As stratigraphic subsidence became progressively decoupled from active fault control, a single, unified subsidence center developed, resulting in a broad, basin-wide depositional framework. Concurrently, source areas advanced and overall relief was reduced, producing a smoother paleotopography. Sedimentation became dominated by large-scale braided-river delta systems prograding along both the short and long axes of the basin. Only limited, shallow-lake deposits were intermittently preserved within the central part of the depression, reflecting reduced accommodation and widespread sediment infilling during rift contraction. During this stage, a gently sloping–braided river delta source–sink system was mainly developed (Fig 17d).

It is noteworthy that, when compared with the East African Rift System, a representative example of a typical continental rift basin, the tectonic evolution of the East African Rift began in the late Eocene (~40 Ma) and was driven by upwelling of the African superplume, which resulted in regional doming. During the Miocene (~22−10 Ma), accelerated uplift of the Kenyan Dome triggered the staged development of the tripartite rift system, comprising the Ethiopian Main Rift, the Western Rift, and the Eastern Rift. The response of the S2S system to this tectonic and geomorphic differentiation exhibits several distinctive characteristics. First, sediment source areas were primarily controlled by rift shoulders, with provenance dominated by Precambrian basement metamorphic and volcanic rocks exposed along the uplifted rift flanks. These source areas supplied short-range, high-energy alluvial fan systems that accumulated along steep rift escarpments. Second, sediment transport pathways varied markedly between rift segments: axial river systems dominated sediment delivery in the Western Rift basins (e.g., Lake Tanganyika), whereas the Eastern Rift (e.g., the Turkana Basin) was characterized by lateral, seasonally active rivers and mixed volcanic-clastic sediment sources. Third, sedimentary sinks were governed by asymmetric subsidence within semi-graben structures, resulting in high-frequency alternations of lacustrine mudstones

and fan-delta sand bodies. Volcanic event layers, such as tuffs, commonly served as regionally correlatable isochronous markers. Overall, the S2S structural evolution of the East African Rift System shows strong conceptual similarities to that of the Lishu Fault Depression. This comparison suggests that the S2S evolution documented in the Lishu Fault Depression is broadly consistent with the development patterns observed in typical continental rift basins worldwide, reinforcing the broader applicability of the proposed multiphase rift S2S model.

## Quantitative relationship of source-sink system

Quantifying S2S relationships remains one of the fundamental challenges in rift-basin research. Despite extensive investigations of modern S2S systems and the development of numerous conceptual and numerical models [43–44], quantitative constraints linking upstream sediment-routing networks to downstream depositional architectures remain limited. In this study, we perform a semi-quantitative analysis of key morphometric parameters of paleochannels within the source-area transport network, including channel width, channel depth, width-depth ratio, and cross-sectional area. These parameters are systematically compared with the geometrical characteristics of fan bodies in the depositional domain (Table 2). The resulting quantitative relationships between valley morphology and topographic metrics reveal distinct, stage-dependent scaling behaviors throughout rift evolution. Regression analyses demonstrate systematic variations in correlation strength, with coefficients of determination ($R^2$) ranging from 0.55 to 0.90 across the three major extensional phases, indicating progressive changes in S2S coupling efficiency during multiphase rifting.

Initial extension stage (blue circles). Valley width exhibits the strongest correlation with topographic parameters ($R^2 > 0.85$), highlighting the dominant structural control exerted by early fault propagation. Steep regression slopes (>1.2) indicate a high sensitivity of channel geometry to initial rift segmentation, reflecting strong tectonic forcing on sediment-routing systems during the onset of extension. Early intense extension stage (orange squares). Valley depth and cross-sectional area display enhanced scaling relationships ($R^2 = 0.75 \pm 0.05$), suggesting tighter coupling between fault-controlled subsidence and sediment accommodation. Regression slopes approach unity (0.95–1.05), marking a transition toward quasi-equilibrium geomorphic conditions in which tectonic forcing and surface processes exert comparable influence. Late intense extension stage (yellow triangles). Correlations among all morphometric parameters weaken ($R^2 = 0.55–0.65$), and regression slopes decrease (<0.8). This attenuation reflects progressive overprinting of tectonic signals by surface processes, as erosion, drainage reorganization, and climate-driven variability become increasingly influential during advanced rift evolution. A fundamental temporal trend emerges in which correlation strength systematically decays from valley width to depth to cross-sectional area, indicating progressive tectono-geomorphic decoupling during rift maturation. Nevertheless, all regressions remain statistically significant ($p < 0.01$), demonstrating that rift-valley morphogenesis retains an inherent scale dependency despite shifts in dominant controls, from strongly fault-governed processes in the early stages to surface-process-dominated dynamics in later stages (Fig 18). It should be noted that the sand-transport parameters quantified in this study do not explicitly incorporate the denudational state of source areas or paleoclimatic variations. Consequently, the results represent statistically robust relationships derived from static morphometric measurements and should be interpreted as reference constraints rather than fully coupled, process-based S2S reconstructions. Overall, the scale of depositional areas in multi-stage rift sedimentary systems is closely related to the characteristics of transport pathways, and is also partly controlled by the rift evolutionary stage. During the early rifting stage, deformation is dominated by lateral extension, and basin stretching is mainly controlled by fault activity, resulting in proximal deposition. Under conditions of stronger fault activity, the areal extent of fan systems tends to be relatively small. In contrast, during the intense rifting stage, sediment supply is sufficient and fault activity is further enhanced, leading to a significant increase in the axial extent of depositional systems. Therefore, the sedimentary area within source-to-sink systems is to a large extent governed by the tectonic phases of the rift basin, while fault activity intensity and stress orientation are the key factors controlling the architecture of the source-to-sink system.

**Table 2. Semi-quantitative characterization of marginal sediment transport channels and depositional fan sizes in the Lishu Fault Depression, southeastern Songliao Basin.**

| Layer | No. | Type | Width/km | Depth/km | Width-Depth Ratio | Cross-Sectional Area/km² | Fan body area/km² |
|---|---|---|---|---|---|---|---|
| Huoshiling Formation | V1 | U | 3.47 | 1.20 | 2.89 | 2.08 | 7.08 |
| | V2 | U | 3.10 | 1.05 | 2.95 | 1.63 | 5.60 |
| | V3 | V | 0.99 | 0.52 | 1.91 | 0.26 | 3.79 |
| | V4 | V | 0.89 | 0.50 | 1.80 | 0.22 | 3.82 |
| | V5 | W | 1.39 | 0.25 | 5.56 | 0.17 | 6.30 |
| | V6 | U | 1.00 | 0.43 | 2.33 | 0.22 | 6.28 |
| | V7 | U | 2.27 | 0.46 | 4.97 | 0.52 | 13.40 |
| | V8 | U | 1.83 | 0.52 | 3.53 | 0.47 | 13.90 |
| | V9 | U | 4.78 | 1.00 | 4.78 | 2.39 | 40.00 |
| | V10 | V | 1.10 | 0.65 | 1.69 | 0.36 | 13.36 |
| | V11 | V | 1.30 | 0.48 | 2.71 | 0.31 | 11.88 |
| | V12 | U | 3.54 | 0.85 | 4.18 | 1.50 | 15.99 |
| | V13 | U | 5.44 | 0.96 | 5.67 | 2.61 | 21.45 |
| | V14 | V | 1.35 | 0.31 | 4.35 | 0.21 | 4.38 |
| | V15 | U | 1.35 | 0.27 | 5.00 | 0.18 | 2.20 |
| | V16 | W | 1.16 | 0.17 | 6.96 | 0.10 | 3.20 |
| | V17 | U | 2.43 | 2.18 | 1.12 | 2.64 | 8.17 |
| | V18 | U | 0.60 | 0.10 | 5.95 | 0.03 | 5.40 |
| Shahezi Formation | V19 | V | 0.73 | 0.31 | 2.37 | 0.11 | 5.60 |
| | V20 | V | 0.96 | 0.38 | 2.55 | 0.18 | 2.80 |
| | V21 | U | 2.39 | 0.86 | 2.78 | 1.02 | 2.10 |
| | V22 | V | 0.74 | 0.20 | 3.69 | 0.07 | 1.00 |
| | V23 | V | 1.00 | 0.28 | 3.65 | 0.14 | 1.39 |
| | V24 | W | 1.21 | 0.20 | 6.03 | 0.12 | 1.56 |
| | V25 | W | 1.60 | 0.25 | 6.30 | 0.20 | / |
| | V26 | W | 2.26 | 0.21 | 10.70 | 0.24 | / |
| | V27 | V | 0.57 | 0.24 | 2.37 | 0.07 | 24.67 |
| | V28 | W | 1.86 | 0.23 | 8.25 | 0.21 | 34.30 |
| | V29 | U | 2.13 | 0.76 | 2.82 | 0.80 | 11.03 |
| | V30 | V | 0.55 | 0.22 | 2.48 | 0.06 | 21.64 |
| | V31 | V | 0.76 | 0.15 | 5.04 | 0.06 | 11.14 |
| | V32 | U | 0.88 | 0.28 | 3.12 | 0.12 | 23.50 |
| | V33 | U | 2.69 | 1.05 | 2.56 | 1.41 | 27.34 |
| | V34 | V | 1.00 | 0.32 | 3.11 | 0.16 | 4.08 |
| | V35 | V | 1.19 | 0.36 | 3.34 | 0.21 | 3.62 |
| | V36 | U | 0.94 | 0.25 | 3.74 | 0.12 | 9.57 |
| | V37 | U | 1.32 | 0.26 | 5.07 | 0.17 | 5.46 |
| | V38 | Fault Trough | 1.34 | 0.39 | 3.44 | 0.26 | 13.20 |
| | V39 | Fault Trough | 1.08 | 0.45 | 2.40 | 0.24 | 10.16 |
| | V40 | Fault Trough | 0.90 | 0.40 | 2.24 | 0.18 | 12.16 |

*(Continued)*

**Table 2.** (Continued)

| Layer | No. | Type | Width/km | Depth/km | Width-Depth Ratio | Cross-Sectional Area/km² | Fan body area/km² |
|---|---|---|---|---|---|---|---|
| Yingcheng Formation | v41 | Fault Trough | 2.54 | 0.92 | 2.76 | 1.17 | 19.50 |
| | v42 | U | 2.35 | 0.62 | 3.80 | 0.73 | 6.71 |
| | v43 | W | 2.60 | 0.30 | 8.58 | 0.39 | 6.74 |
| | v44 | U | 9.93 | 0.90 | 11.03 | 4.47 | 17.90 |
| | v45 | V | 3.74 | 1.18 | 3.17 | 2.21 | 9.50 |
| | v46 | V | 1.44 | 0.42 | 3.46 | 0.30 | 7.17 |
| | v47 | V | 1.50 | 0.69 | 2.16 | 0.52 | 6.58 |
| | v48 | U | 1.46 | 0.24 | 6.02 | 0.18 | 5.68 |
| | v49 | U | 1.25 | 0.60 | 2.08 | 0.37 | / |
| | v50 | U | 2.76 | 0.30 | 9.20 | 0.41 | / |
| | v51 | U | 1.49 | 0.25 | 5.96 | 0.19 | / |
| | v52 | U | 2.80 | 0.36 | 7.78 | 0.50 | / |
| | v53 | U | 1.40 | 0.18 | 7.78 | 0.13 | 18.80 |
| | v54 | U | 2.35 | 0.26 | 8.94 | 0.31 | 20.76 |
| | v55 | U | 1.00 | 0.18 | 5.56 | 0.09 | 16.05 |
| | v56 | U | 1.84 | 0.37 | 4.97 | 0.34 | 15.90 |
| | v57 | Fault Trough | 1.14 | 0.26 | 4.38 | 0.15 | / |
| | v58 | Fault Trough | 1.02 | 0.34 | 3.01 | 0.17 | / |
| | v59 | Fault Trough | 0.69 | 0.20 | 3.47 | 0.07 | / |
| | v60 | Fault Trough | 2.14 | 0.42 | 5.10 | 0.45 | / |
| | v61 | Fault Trough | 0.84 | 0.25 | 3.37 | 0.11 | / |
| | v62 | U | 2.48 | 0.43 | 5.76 | 0.53 | / |
| Den-glouku Formation | v63 | U | 2.37 | 0.31 | 7.78 | 0.36 | / |
| | v64 | U | 1.90 | 0.41 | 4.62 | 0.39 | / |
| | v65 | U | 1.10 | 0.37 | 2.97 | 0.20 | / |
| | v66 | U | 1.45 | 0.38 | 3.84 | 0.27 | / |
| | v67 | V | 3.45 | 1.55 | 2.22 | 2.67 | / |
| | v68 | U | 2.18 | 0.53 | 4.11 | 0.58 | / |
| | v69 | U | 1.08 | 0.28 | 3.85 | 0.15 | / |
| | v70 | U | 0.74 | 0.28 | 2.64 | 0.10 | / |
| | v71 | U | 1.20 | 0.16 | 7.59 | 0.09 | / |
| | v72 | U | 1.22 | 0.13 | 9.38 | 0.08 | / |
| | v73 | U | 0.57 | 0.16 | 3.59 | 0.05 | / |
| | v74 | W | 1.53 | 0.15 | 10.01 | 0.12 | / |
| | v75 | U | 1.70 | 0.19 | 8.85 | 0.16 | / |
| | v76 | U | 0.88 | 0.23 | 3.83 | 0.10 | / |
| | v77 | U | 1.45 | 0.27 | 5.31 | 0.20 | / |
| | v78 | U | 1.02 | 0.30 | 3.37 | 0.15 | / |
| | v79 | U | 1.56 | 0.32 | 4.93 | 0.25 | / |
| | v80 | U | 0.83 | 0.27 | 3.03 | 0.11 | / |
| | v81 | U | 0.88 | 0.18 | 5.02 | 0.08 | / |
| | v82 | U | 0.86 | 0.27 | 3.20 | 0.12 | / |
| | v83 | Fault Trough | 0.66 | 0.35 | 1.88 | 0.12 | / |
| | v84 | Fault Trough | 1.08 | 0.33 | 3.27 | 0.18 | / |

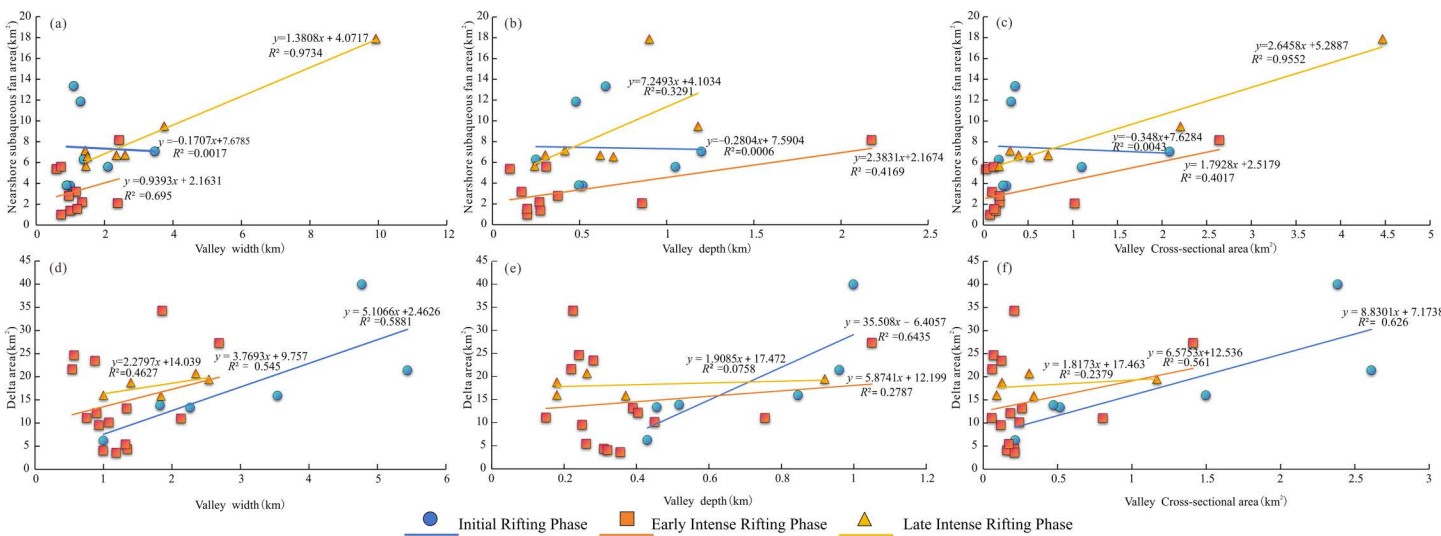

**Fig 18. Geomorphic relationship between sediment transport channel parameters and delta/fan complexes in the Lishu Fault Depression.**

## Conclusions

This study clarifies the coupling mechanisms and evolutionary framework of the S2S system in the Lishu Fault Depression during multiphase rifting, providing an integrated perspective on how tectonic processes, sediment routing, and depositional responses co-evolved through successive stages of rift development.

SQ1-SQ6 were identified, spanning the Huoshiling to Yingcheng formations. Boundary faults, particularly those in the Sangshutai area, display pronounced spatial variability in activity, with the central fault segment experiencing the highest displacement rates during the late rifting stage. This differential fault activity exerted a fundamental control on the generation and distribution of accommodation space and, in turn, strongly influenced sediment-routing pathways.

Sediment supply to the basin was predominantly multisourced. Erosional valleys, structural transfer zones, and parallel fault-step belts served as the principal controls on sediment transport and deposition. Valley systems focused sand delivery from source areas, transfer zones facilitated drainage reorganization and sediment redistribution, and fault-step belts modulated accommodation space in a stepwise fashion, collectively directing sand bodies toward basinward depocenters.

During the initial rifting stage, sediment supply was partitioned, leading to the development of steep-slope nearshore subaqueous fans and a differentiated source-to-sink coupling pattern of steep slope–fan delta systems. In the early stage of intense rifting, sediment supply became convergent, and the dominant source-to-sink coupling patterns included steep slope–transfer zone–fan delta systems, axial slope–fan delta systems, and eastern gentle slope parallel fault-step systems. In the late stage, axial drainage systems became dominant, primarily forming axial slope–braided river delta coupling systems. During the late rifting stage, axial sediment supply was further strengthened, and the system was characterized by the development of gentle slope–braided river delta source-to-sink coupling patterns.

Statistical analyses further demonstrate that the scale of nearshore subaqueous fans and deltaic deposits is positively correlated with key geomorphic parameters of the sediment-routing system. These relationships provide quantitative evidence linking tectonic forcing to sediment transport pathways and depositional responses, thereby reinforcing the S2S coupling framework proposed in this study.

## Supporting information

**S1 Table. Heavy mineral data and Q-F-L data (K1h-K1d).** The data represents a summary of heavy minerals from the Huoshiling Formation-Denglouku Formation and a statistical summary of quartz, feldspar, and lithic fragments. A pie chart has been plotted for the heavy minerals, and a ternary diagram has been drawn to show the content of quartz, feldspar, and lithic fragments.
(ZIP)

## Acknowledgments

We would like to express our sincere gratitude to Engineer Xiao Meng from the Northeast Petroleum Bureau of Sinopec for his valuable support. We also thank the editors and anonymous reviewers for their insightful comments and constructive suggestions, which greatly improved this manuscript. Our appreciation is further extended to China University of Petroleum (Beijing) for providing an excellent research platform.

## Author contributions

**Conceptualization:** Ke Wang.

**Data curation:** Jingchi Yan.

**Formal analysis:** Jingchi Yan.

**Funding acquisition:** Yong Zhou.

**Investigation:** Jingchi Yan.

**Methodology:** Yong Zhou.

**Resources:** Yuejie Zhang.

**Software:** Yuejie Zhang.

**Supervision:** Yuejie Zhang.

**Visualization:** Yuejie Zhang.

**Writing – original draft:** Ke Wang.

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
