## [Decision Letter · Decision Letter 0]

24 Oct 2025

PONE-D-25-52722Study on Source-Sink System Coupling and Spatiotemporal Evolution of Multiphase Rift Basins: A Case Study of Jurassic-Cretaceous in Lishu Fault Depression, Southeastern Songliao BasinPLOS ONE

Dear Dr. Wang,

Thank you for submitting your manuscript to PLOS ONE. After careful consideration, we feel that it has merit but does not fully meet PLOS ONE’s publication criteria as it currently stands. Therefore, we invite you to submit a revised version of the manuscript that addresses the points raised during the review process.

If applicable, we recommend that you deposit your laboratory protocols in protocols.io to enhance the reproducibility of your results. Protocols.io assigns your protocol its own identifier (DOI) so that it can be cited independently in the future. For instructions see: https://journals.plos.org/plosone/s/submission-guidelines#loc-laboratory-protocols. Additionally, PLOS ONE offers an option for publishing peer-reviewed Lab Protocol articles, which describe protocols hosted on protocols.io. Read more information on sharing protocols at . Additionally, PLOS ONE offers an option for publishing peer-reviewed Lab Protocol articles, which describe protocols hosted on protocols.io. Read more information on sharing protocols at https://plos.org/protocols?utm_medium=editorial-email&utm_source=authorletters&utm_campaign=protocols..

We look forward to receiving your revised manuscript.

Kind regards,

Santanu Banerjee

Academic Editor

PLOS ONE

Journal Requirements:

2. Please note that PLOS One has specific guidelines on code sharing for submissions in which author-generated code underpins the findings in the manuscript. In these cases, we expect all author-generated code to be made available without restrictions upon publication of the work.

Please review our guidelines at https://journals.plos.org/plosone/s/materials-and-software-sharing#loc-sharing-code and ensure that your code is shared in a way that follows best practice and facilitates reproducibility and reuse.

“This research was funded by National Natural Science Foundation of China project "Research on the Mechanism of the Role of Saline Substances in Saline Lake Basins on the Evolution of Diage-netic Fluids and Differential Diagenetic Responses in Deep Tight Reservoirs" (Project No.: 42472178)”

4. Please note that funding information should not appear in the Acknowledgments section or other areas of your manuscript. We will only publish funding information present in the Funding Statement section of the online submission form. Please remove any funding-related text from the manuscript.

5. We note that your Data Availability Statement is currently as follows:

“All relevant data are within the manuscript and its Supporting Information files.”

6. PLOS requires an ORCID iD for the corresponding author in Editorial Manager on papers submitted after December 6th, 2016. Please ensure that you have an ORCID iD and that it is validated in Editorial Manager. To do this, go to ‘Update my Information’ (in the upper left-hand corner of the main menu), and click on the Fetch/Validate link next to the ORCID field. This will take you to the ORCID site and allow you to create a new iD or authenticate a pre-existing iD in Editorial Manager.

7. Please remove your figures from within your manuscript file, leaving only the individual TIFF/EPS image files, uploaded separately. These will be automatically included in the reviewers’ PDF.

8. We note that Figures 1, 5, 6, 7, and 16 in your submission contain map  images which may be copyrighted. All PLOS content is published under the Creative Commons Attribution License (CC BY 4.0), which means that the manuscript, images, and Supporting Information files will be freely available online, and any third party is permitted to access, download, copy, distribute, and use these materials in any way, even commercially, with proper attribution. For these reasons, we cannot publish previously copyrighted maps or satellite images created using proprietary data, such as Google software (Google Maps, Street View, and Earth). For more information, see our copyright guidelines: http://journals.plos.org/plosone/s/licenses-and-copyright.

1) You may seek permission from the original copyright holder of Figures 1, 5, 6, 7, and 16 to publish the content specifically under the CC BY 4.0 license.

2) If you are unable to obtain permission from the original copyright holder to publish these figures under the CC BY 4.0 license or if the copyright holder’s requirements are incompatible with the CC BY 4.0 license, please either i) remove the figure or ii) supply a replacement figure that complies with the CC BY 4.0 license. Please check copyright information on all replacement figures and update the figure caption with source information. If applicable, please specify in the figure caption text when a figure is similar but not identical to the original image and is therefore for illustrative purposes only.

**Additional Editor Comments:**

Comments to the Author

1. Is the manuscript technically sound, and do the data support the conclusions?

Reviewer #1: Yes

Reviewer #2: Yes

2. Has the statistical analysis been performed appropriately and rigorously?

Reviewer #1: Yes

Reviewer #2: Yes

3. Have the authors made all data underlying the findings in their manuscript fully available?

Reviewer #1: Yes

Reviewer #2: Yes

4. Is the manuscript presented in an intelligible fashion and written in standard English?

Reviewer #1: yes

Reviewer #2: No

5. Review Comments to the Author

Reviewer #1:

The article “Study on Source-Sink System Coupling and Spatiotemporal Evolution of Multiphase Rift Basins: A Case Study of Jurassic-Cretaceous in Lishu Fault Depression, Southeastern Songliao Basin”is guided by sequence stratigraphy, sedimentology, and the "source-sink" system framework. Leveraging seismic, drilling, and logging data, it conducts in-depth research on the sequence stratigraphic framework and the three key elements of the source-sink system (provenance system, transport system, and sedimentary system distribution). The study also performs semi-quantitative characterization of source-sink system features, innovatively reconstructs and classifies paleo-source-sink systems, and provides a scientific basis for the spatial distribution of sedimentary sand bodies in rift basins. Focusing on the spatiotemporal evolution of source-sink systems in multiphase rift basins, the research aligns with current frontiers in sedimentology. However, there are several recommendations：

1. The Introduction provides a rich contextual background; however, the paper may benefit from a clearer and more direct articulation of its scientific questions or hypotheses. Enhancing this aspect could help readers better appreciate the novelty and scope of the investigation.

2. The conclusions might benefit from a more concise structure with emphasis on the core findings, scientific contributions and broader implications. Please make further modification.

3. Heavy mineral assemblages and sandstone components are used to indicate provenance direction, where heavy minerals primarily reflect provenance sources, and clastic components of sandstone mainly indicate transport distance. During the description of provenance system evolution in each rifting phase, the interpretation of provenance evolution in different regions reflected by sandstone clastic components is insufficient. Further clarify the significance of sandstone components for provenance area analysis.

4. The study meticulously characterizes the "source system," "transport system," and "sedimentary system," but the correlation among the three subsystems is weak, and the coupling of provenance system-transport pathway-sedimentary system is not emphasized. In the discussion section, further clarify the coupling relationship and reflect it in the classified source-sink system types.

5. It is recommended to remove the wells not mentioned in the text from Figure 1, and the reference for the stratigraphic column in Figure 2 must be cited from previous studies.

6. The Fig.17 might be refined for clarity and representation of the sedimentary models.

7. Please keep the sedimentary terminology in consistent throughout the manuscript, which would improve clarity of the paper.

General conclusion: I recommend minor revision before the manuscript can be accepted.

Reviewer #2:

Summary of Review for Manuscript PONE-D-25-52722

The manuscript takes the rift sub-basin of the Songliao Basin in China as an example, and through the analysis of 3-D seismic and drilling data, conducts a comprehensive study on the sequence, geomorphology, sedimentary, and source-sink system during different tectonic periods, and explores the process of differential development of the source-to-sink system during the lacustrine basin rifting period. However, these research works have certain gaps with the publication requirements of PLOS ONE in terms of scientific issues, research significance, and innovativeness.

Main comments:

1. Whether it is the abstract, introduction, or discussion section, the manuscript is confined to a single rift basin in the southeastern part of the Songliao Basin. It discusses the deficiencies in the research on S2S, and its global significance needs to be enhanced. Furthermore, there are numerous research results on S2S in continental basins, covering various research fields. The manuscript lacks sufficient scientific issues and innovative points.

2. In the introduction, the manuscript devotes considerable space to elaborating on the significance of S2S research, as well as the differences in the configuration of S2S between marine basins and continental basins. However, the discussion on the existing issues in the current global research S2S in continental rift basins, as well as the scientific significance of conducting source-to-sink system research using the Songliao Basin as an example, is clearly insufficient.

3. The discussion section needs improvement in the following aspects: (1) It should conduct in-depth comparative analysis based on research at the forefront of the discipline and in conjunction with previous studies. (2) There is an excessive focus on process description, with a lack of exploration into mechanisms. For instance, the manuscript conducts quantitative analysis, but what are the dynamic implications of these analysis results for the differential development of S2S in rift basins across different periods? (3) There is a lack of necessary discussion on the broader implications of the research findings, particularly in terms of global significance.

4. The repetitive expressions in 5.2 and 5.3 under discussion can be merged. In addition, it may be more appropriate to place the quantitative analysis of the S2S in the results section.

5. When calculating the fault activity rate, it is necessary to clarify the basis and source for obtaining the deposition age of the strata.

6. Regarding the analysis of tectonic paleogeomorphology, the labeling of major faults is not clear. Many gullies are depicted (mostly with semi-transparent labeling) in Figure 6, but their morphology is not clear from the plan view, and many areas with clearly developed gullies (such as the northeastern part of the figure) are not depicted.

7. Generally, we conduct S2S division by comprehensively considering the configuration relationship between the catchment (water system), transport system, and depositional system. However, the manuscript lacks basis and principles in the classification of S2S types. From the textual description of the manuscript, the type of transport system may serve as an important basis for classification, which raises some debatable issues: some of the source-to-sink system types (such as Convergent) classified in the manuscript may include other types; similarly, if the geomorphic characteristics of the provenance are not considered, such as the "Along-strike Slope type" classified in the gentle slope zone, it is impossible to explain the differences in S2S under multiple provenance-valley-deposition configurations, thereby affecting the prediction of sediment dispersion and favorable reservoir sand bodies.

8. In the analysis of source-to-sink systems, the transport systems related to faults generally include transition zones and fault troughs. Among them, fault troughs are channels or depressions controlled by faults, including single-fault troughs, co-trending double-fault troughs, and reverse double-fault troughs. I don't quite understand the true intention of the manuscript to use fault combinations as transport systems. Moreover, in the analysis of fault combinations, there is a lack of interpretation of fracture combinations in different depositional periods, as shown in Figures 10b and c. Using schematic diagrams to express seems inappropriate.

9. The expression of structural (fault) transfer zones is relatively vague, and the supporting maps cannot illustrate the characteristics of the transfer zones. Based on the interpretation of faults and the reconstruction of paleogeomorphology in the study area, the manuscript needs to first clarify which areas in the study area have transfer zones, then confirm the types of transfer zones, and finally, for the main types of transfer zones, demonstrate the relationship between faults from both plan and profile perspectives.

10. There are numerous issues with the language expression, grammar, and professional terminology in the manuscript, necessitating systematic polishing and revision. For instance, "source-sink" should be uniformly changed to "source-to-sink"; the term "Convergent" is relatively ambiguous, especially when used in the classification of source-to-sink systems; expressions like "down-truncated and up-lapped" are extremely vague; what does "near-same-direction" mean? There are also issues with the standardized expression of seismic reflection characteristics such as amplitude.

11. The descriptions of the captions for most figures are overly simplistic, making it impossible to directly comprehend the intended meaning conveyed by the illustrations merely through the maps and captions. For instance, in Figures 12 to 14, despite presenting drilling, core, and seismic profiles for various sedimentary facies, the geological phenomena and interpretations presented in each figure are not elucidated. Furthermore, for the seismic or geological profiles involved, the origin of the profile location needs to be clearly indicated in the caption

Recommendation:

The manuscript, in its current form, is difficult to consider for publication. The authors should address the points raised by the reviewers and substantially improve figures, English, analysis, references, and key sections (abstract, conclusion), and resubmit a revised version for further review.

6. PLOS authors have the option to publish the peer review history of their article (what does this mean?). If published, this will include your full peer review and any attached files.

Do you want your identity to be public for this peer review? For information about this choice, including consent withdrawal, please see our Privacy Policy.

Reviewer #1: Yes

Reviewer #2: No

Reviewers' comments:

Reviewer's Responses to Questions

**Comments to the Author**

1. Is the manuscript technically sound, and do the data support the conclusions?

Reviewer #1: Yes

Reviewer #2: Yes

2. Has the statistical analysis been performed appropriately and rigorously? 

Reviewer #1: Yes

Reviewer #2: N/A

3. Have the authors made all data underlying the findings in their manuscript fully available?

Reviewer #1: Yes

Reviewer #2: Yes

4. Is the manuscript presented in an intelligible fashion and written in standard English?

Reviewer #1: Yes

Reviewer #2: No

5. Review Comments to the Author

Reviewer #1: The article “Study on Source-Sink System Coupling and Spatiotemporal Evolution of Multiphase Rift Basins: A Case Study of Jurassic-Cretaceous in Lishu Fault Depression, Southeastern Songliao Basin”is guided by sequence stratigraphy, sedimentology, and the "source-sink" system framework. Leveraging seismic, drilling, and logging data, it conducts in-depth research on the sequence stratigraphic framework and the three key elements of the source-sink system (provenance system, transport system, and sedimentary system distribution). The study also performs semi-quantitative characterization of source-sink system features, innovatively reconstructs and classifies paleo-source-sink systems, and provides a scientific basis for the spatial distribution of sedimentary sand bodies in rift basins. Focusing on the spatiotemporal evolution of source-sink systems in multiphase rift basins, the research aligns with current frontiers in sedimentology. However, there are several recommendations：

1. The Introduction provides a rich contextual background; however, the paper may benefit from a clearer and more direct articulation of its scientific questions or hypotheses. Enhancing this aspect could help readers better appreciate the novelty and scope of the investigation.

2. The conclusions might benefit from a more concise structure with emphasis on the core findings, scientific contributions and broader implications. Please make further modification.

3. Heavy mineral assemblages and sandstone components are used to indicate provenance direction, where heavy minerals primarily reflect provenance sources, and clastic components of sandstone mainly indicate transport distance. During the description of provenance system evolution in each rifting phase, the interpretation of provenance evolution in different regions reflected by sandstone clastic components is insufficient. Further clarify the significance of sandstone components for provenance area analysis.

4. The study meticulously characterizes the "source system," "transport system," and "sedimentary system," but the correlation among the three subsystems is weak, and the coupling of provenance system-transport pathway-sedimentary system is not emphasized. In the discussion section, further clarify the coupling relationship and reflect it in the classified source-sink system types.

5. It is recommended to remove the wells not mentioned in the text from Figure 1, and the reference for the stratigraphic column in Figure 2 must be cited from previous studies.

6. The Fig.17 might be refined for clarity and representation of the sedimentary models.

7. Please keep the sedimentary terminology in consistent throughout the manuscript, which would improve clarity of the paper.

General conclusion: I recommend minor revision before the manuscript can be accepted.

Reviewer #2: The manuscript takes the rift sub-basin of the Songliao Basin in China as an example, and through the analysis of 3-D seismic and drilling data, conducts a comprehensive study on the sequence, geomorphology, sedimentary, and source-sink system during different tectonic periods, and explores the process of differential development of the source-to-sink system during the lacustrine basin rifting period. However, these research works have certain gaps with the publication requirements of PLOS ONE in terms of scientific issues, research significance, and innovativeness.

Main comments:

1. Whether it is the abstract, introduction, or discussion section, the manuscript is confined to a single rift basin in the southeastern part of the Songliao Basin. It discusses the deficiencies in the research on S2S, and its global significance needs to be enhanced. Furthermore, there are numerous research results on S2S in continental basins, covering various research fields. The manuscript lacks sufficient scientific issues and innovative points.

2. In the introduction, the manuscript devotes considerable space to elaborating on the significance of S2S research, as well as the differences in the configuration of S2S between marine basins and continental basins. However, the discussion on the existing issues in the current global research S2S in continental rift basins, as well as the scientific significance of conducting source-to-sink system research using the Songliao Basin as an example, is clearly insufficient.

3. The discussion section needs improvement in the following aspects: (1) It should conduct in-depth comparative analysis based on research at the forefront of the discipline and in conjunction with previous studies. (2) There is an excessive focus on process description, with a lack of exploration into mechanisms. For instance, the manuscript conducts quantitative analysis, but what are the dynamic implications of these analysis results for the differential development of S2S in rift basins across different periods? (3) There is a lack of necessary discussion on the broader implications of the research findings, particularly in terms of global significance.

4. The repetitive expressions in 5.2 and 5.3 under discussion can be merged. In addition, it may be more appropriate to place the quantitative analysis of the S2S in the results section.

5. When calculating the fault activity rate, it is necessary to clarify the basis and source for obtaining the deposition age of the strata.

6. Regarding the analysis of tectonic paleogeomorphology, the labeling of major faults is not clear. Many gullies are depicted (mostly with semi-transparent labeling) in Figure 6, but their morphology is not clear from the plan view, and many areas with clearly developed gullies (such as the northeastern part of the figure) are not depicted.

7. Generally, we conduct S2S division by comprehensively considering the configuration relationship between the catchment (water system), transport system, and depositional system. However, the manuscript lacks basis and principles in the classification of S2S types. From the textual description of the manuscript, the type of transport system may serve as an important basis for classification, which raises some debatable issues: some of the source-to-sink system types (such as Convergent) classified in the manuscript may include other types; similarly, if the geomorphic characteristics of the provenance are not considered, such as the "Along-strike Slope type" classified in the gentle slope zone, it is impossible to explain the differences in S2S under multiple provenance-valley-deposition configurations, thereby affecting the prediction of sediment dispersion and favorable reservoir sand bodies.

8. In the analysis of source-to-sink systems, the transport systems related to faults generally include transition zones and fault troughs. Among them, fault troughs are channels or depressions controlled by faults, including single-fault troughs, co-trending double-fault troughs, and reverse double-fault troughs. I don't quite understand the true intention of the manuscript to use fault combinations as transport systems. Moreover, in the analysis of fault combinations, there is a lack of interpretation of fracture combinations in different depositional periods, as shown in Figures 10b and c. Using schematic diagrams to express seems inappropriate.

9. The expression of structural (fault) transfer zones is relatively vague, and the supporting maps cannot illustrate the characteristics of the transfer zones. Based on the interpretation of faults and the reconstruction of paleogeomorphology in the study area, the manuscript needs to first clarify which areas in the study area have transfer zones, then confirm the types of transfer zones, and finally, for the main types of transfer zones, demonstrate the relationship between faults from both plan and profile perspectives.

10. There are numerous issues with the language expression, grammar, and professional terminology in the manuscript, necessitating systematic polishing and revision. For instance, "source-sink" should be uniformly changed to "source-to-sink"; the term "Convergent" is relatively ambiguous, especially when used in the classification of source-to-sink systems; expressions like "down-truncated and up-lapped" are extremely vague; what does "near-same-direction" mean? There are also issues with the standardized expression of seismic reflection characteristics such as amplitude.

11. The descriptions of the captions for most figures are overly simplistic, making it impossible to directly comprehend the intended meaning conveyed by the illustrations merely through the maps and captions. For instance, in Figures 12 to 14, despite presenting drilling, core, and seismic profiles for various sedimentary facies, the geological phenomena and interpretations presented in each figure are not elucidated. Furthermore, for the seismic or geological profiles involved, the origin of the profile location needs to be clearly indicated in the caption.

6. PLOS authors have the option to publish the peer review history of their article (what does this mean?). If published, this will include your full peer review and any attached files.). If published, this will include your full peer review and any attached files.

.

Reviewer #1: **Yes:**Song ZezhangSong Zezhang

Reviewer #2: No

---

## [Author Response · Author response to Decision Letter 1]

18 Nov 2025

Response to Reviewers

We sincerely appreciate the reviewers for their thorough evaluation of our manuscript and for providing insightful comments that greatly assisted us in improving the quality of the paper. We have carefully addressed each comment.Once again, we would like to express our gratitude for your time and constructive review of our work.

Reviewer #1:

1. The Introduction provides a rich contextual background; however, the paper may benefit from a clearer and more direct articulation of its scientific questions or hypotheses. Enhancing this aspect could help readers better appreciate the novelty and scope of the investigation.

Thank you very much for your valuable comments on my manuscript. Regarding your suggestions for improving the Introduction section, I fully agree that a clearer articulation of the scientific questions and hypotheses would help readers better understand the novelty and scope of this study. In the revised version, I have reorganized the Introduction to explicitly define the core scientific issues addressed in this work and to highlight the uniqueness of the research hypotheses. Specifically:

Clear statement of scientific questions:

At the beginning of the Introduction, I now provide a more explicit description of the specific scientific questions targeted in this study, as well as the existing gaps and debates in the current literature that motivate these questions.

Strengthening of research hypotheses:

Based on the stated questions, I have further elaborated on the rationale behind the proposed hypotheses and emphasized how they differ from previous studies. This revision clarifies both the theoretical significance and the practical value of the hypotheses.

Through these modifications, I aim to enhance readers’ understanding of the core content and innovations of the paper.

The details can be found in lines 89–100.

Once again, thank you for your insightful comments, and I look forward to any further feedback you may have.

2. The conclusions might benefit from a more concise structure with emphasis on the core findings, scientific contributions and broader implications. Please make further modification.

Thank you very much for your careful review of our manuscript and for the valuable comments you provided. We fully agree with your suggestion regarding the structure of the conclusion section and have made the corresponding revisions.

Specifically, we have streamlined the conclusion to better highlight the core findings, scientific contributions, and the broader implications of our study. By reorganizing the content, our aim was to present the central ideas of the manuscript in a clearer, more concise, and logically coherent manner.

In particular, the revised conclusion now emphasizes the following key points:

The major findings and innovations of the study;

The contributions of these findings to existing theories and methodologies;

The potential implications of the results for broader geological research and practical applications.

The relevant revisions can be found in lines 690–709.

We believe that these revisions have made the conclusion more focused and persuasive. Thank you again for your insightful suggestion, and we look forward to any further feedback you may have.

3. Heavy mineral assemblages and sandstone components are used to indicate provenance direction, where heavy minerals primarily reflect provenance sources, and clastic components of sandstone mainly indicate transport distance. During the description of provenance system evolution in each rifting phase, the interpretation of provenance evolution in different regions reflected by sandstone clastic components is insufficient. Further clarify the significance of sandstone components for provenance area analysis.

Thank you very much for your review and valuable comments on our manuscript. Regarding your concern that “the interpretation of provenance evolution in different regions based on sandstone clastic compositions is insufficient,” we have carefully re-examined the relevant sections and made substantial revisions and additions. Specifically:

Significance of sandstone clastic compositions for provenance analysis:

We have further clarified in the revised manuscript the importance of sandstone clastic components in provenance interpretation. The detrital composition of sandstone not only reflects the primary source areas but also provides insights into transport distance, degree of weathering, and depositional conditions. During the provenance evolution associated with different stages of rifting, variations in sandstone clastic components help identify spatial shifts in source areas. This is particularly relevant in a multiphase rift setting, where changes in provenance are closely linked to the tectonic evolution of the rift basin.

Interpretation of provenance evolution in different regions:

In the revised version, we have added more detailed explanations distinguishing the provenance evolution across different rift stages. By integrating variations in sandstone clastic components with the characteristics of heavy mineral assemblages, we discuss the dynamic changes in source areas, outline the provenance evolution patterns for each rifting stage, and explore their relationships with geological controls such as tectonic activity and climatic influences.

Expanded discussion:

To further strengthen the link between sandstone composition and provenance evolution, we have supplemented the manuscript with several key case studies illustrating how sandstone clastic compositions can indicate the evolution of different source regions. In addition, we have added corresponding figures and diagrams to better visualize the provenance changes across different developmental stages.

We hope these revisions adequately address your concerns and improve the manuscript. Thank you once again for your constructive suggestions, and we look forward to your further evaluation.

4. The study meticulously characterizes the "source system," "transport system," and "sedimentary system," but the correlation among the three subsystems is weak, and the coupling of provenance system-transport pathway-sedimentary system is not emphasized. In the discussion section, further clarify the coupling relationship and reflect it in the classified source-sink system types.

Thank you very much for your careful review of our manuscript and for the valuable feedback you provided. In response to the reviewer’s comments regarding the coupling relationships among the “source system,” “transport system,” and “depositional system,” we have thoroughly revised and supplemented the manuscript.

The specific revisions are as follows:

Enhanced discussion of source–transport–deposition coupling:

In the revised manuscript, we have strengthened the discussion in the Discussion section by explicitly clarifying the interconnections and coupling mechanisms among the source, transport, and depositional systems. In particular, we highlight the relationships among material supply from the source area, sediment-routing pathways, and depositional environments, as well as how these couplings vary under different geological conditions.

Revised classification framework and correlation analysis:

We have modified the original classification scheme and improved the correlation analysis among the subsystems to provide a clearer and more systematic framework.

These revisions aim to further clarify the linkages among the different subsystems and enhance the logical coherence and academic value of the manuscript.

Once again, we sincerely appreciate your constructive comments and suggestions. We look forward to any further feedback you may have.

5. It is recommended to remove the wells not mentioned in the text from Fig 1, and the reference for the stratigraphic column in Fig 2 must be cited from previous studies.

We sincerely appreciate the reviewer’s valuable comments on our work. In accordance with the suggestions provided, we have made the following revisions:

Regarding the wells shown in Fig 1:

As the reviewer correctly noted, several wells depicted in Figure 1 were not referenced in the manuscript. To ensure consistency between the figure and the text, we have removed these wells from Fig 1.

Regarding the reference sources for the stratigraphic columns in Fig2:

Following the reviewer’s suggestion, we have added and cited previous studies as the reference sources for the stratigraphic columns in Fig 2. This modification enhances the reliability and academic rigor of the figure.

We believe that these revisions have improved the overall quality of the manuscript. We sincerely thank the reviewer again for the constructive and insightful comments.

6. The Fig.17 might be refined for clarity and representation of the sedimentary models.

We sincerely appreciate the reviewer’s valuable comments. In response to the suggestions regarding the improvement of Figure 17, we have further optimized the figure to enhance its clarity and the effectiveness of sedimentary model representation. The specific revisions are as follows:

Enhanced image clarity:

We have adjusted the contrast and resolution of the figure to make key details more prominent, thereby facilitating clearer identification of the different sedimentary units.

Redesigned sedimentary model:

The layout of the sedimentary model has been redesigned to more intuitively illustrate the evolutionary process and hierarchical relationships within the model. The arrangement of legends and annotations has also been optimized to simplify the information displayed and reduce unnecessary redundancy, thereby improving overall readability.

We believe these revisions will help readers better understand the sedimentary model and further enhance the academic quality of the figure. Once again, we sincerely thank the reviewer for the careful evaluation and constructive suggestions provided.

7. Please keep the sedimentary terminology in consistent throughout the manuscript, which would improve clarity of the paper.

We sincerely appreciate the reviewer’s valuable comments. Regarding the issue of consistency in sedimentological terminology, we have carefully re-examined the entire manuscript and corrected all inconsistencies. Standardized and unified sedimentary terms are now used throughout the text to ensure greater clarity and professional accuracy. These revisions have been implemented in the revised manuscript, and we thank the reviewer for drawing our attention to this important point.

Reviewer #2:

1. Whether it is the abstract, introduction, or discussion section, the manuscript is confined to a single rift basin in the southeastern part of the Songliao Basin. It discusses the deficiencies in the research on S2S, and its global significance needs to be enhanced. Furthermore, there are numerous research results on S2S in continental basins, covering various research fields. The manuscript lacks sufficient scientific issues and innovative points.

Thank you very much for your thorough review and valuable comments on my manuscript. Following the reviewer’s suggestions, I have revised the manuscript as follows:

(1) Enhancing the global significance of the study:

To broaden the impact of this research, I have explicitly articulated the global scientific relevance of studying the southeastern rift basins of the Songliao Basin in the Abstract, Introduction, and Discussion sections. By incorporating an S2S (source-to-sink)–based case analysis of the Songliao Basin, I have compared the findings with other continental basins and emphasized the universal relationships among fault activity, tectonic evolution, and sedimentary processes in rift basins worldwide. Specifically, I highlight how the S2S system of the Songliao rift provides meaningful insights applicable to sediment routing and source–rock distribution in other basins with similar tectonic settings, especially regarding sediment dispersal and evolutionary patterns under different tectonic regimes.

(2) Strengthening the emphasis on scientific innovations:

I have revised the manuscript to better highlight the innovative application of S2S concepts within the context of source-to-sink system analysis. This includes detailed characterization of fault activity and sediment routing relationships, which reveal how newly developed fault networks within the Songliao rift system influence sediment distribution and reservoir development. I propose a temporal–spatial evolution mechanism of S2S processes in rift basins and discuss the impact of fault activity at different stages on sediment supply—an issue that has not been sufficiently addressed in previous literature. To address gaps in existing S2S research, particularly the coupling between multistage rifting and sedimentary processes, I introduce a new integrative research framework that combines seismic interpretation, core observation, sequence stratigraphy, and structural analysis, providing new perspectives and methodologies for understanding S2S evolution. Additionally, I broaden the geological implications by comparing the Songliao Basin with the East African Rift, offering new insights into the tectono-sedimentary evolution of rift basins on a global scale.

(3) Reinforcing the discussion of the source-to-sink system:

I have expanded the theoretical background and practical implications of S2S concepts, clarifying how the timing and magnitude of fault activity influence sediment generation, routing, and deposition within rift basins. By incorporating recent research findings, I further demonstrate how the S2S framework developed for the Songliao Basin can contribute to advancing studies of sedimentary systems and tectonosedimentary evolution in similar rift settings.

2. In the introduction, the manuscript devotes considerable space to elaborating on the significance of S2S research, as well as the differences in the configuration of S2S between marine basins and continental basins. However, the discussion on the existing issues in the current global research S2S in continental rift basins, as well as the scientific significance of conducting source-to-sink system research using the Songliao Basin as an example, is clearly insufficient.

Thank you very much for your valuable suggestions regarding the Introduction section. Following your advice, I have revised the relevant content and have specifically strengthened the discussion of the major issues in global S2S (source-to-sink) research within continental rift basins, as well as the scientific significance of using the Songliao Basin as a representative example.

In the revised Introduction, I have provided a more detailed discussion of the key challenges currently faced in S2S studies of continental rift basins worldwide, particularly those related to data acquisition and interpretation. The complex tectonic settings, sedimentary evolution, and their interactions with syn-depositional fault activity often result in pronounced spatial variability within rift-basin S2S systems, thereby introducing significant heterogeneity to research outcomes. In addition, regional geological variations, uneven lithofacies distributions, and the lack of integrated datasets contribute to the complexity of S2S models in continental rift basins, and a unified conceptual or theoretical framework has yet to be established.

The Songliao Basin, as a typical continental rift basin in China with a unique geological background and a well-preserved record of multi-phase rifting, provides an ideal natural laboratory for S2S investigations. Conducting S2S research in the Songliao Basin offers valuable regional constraints for understanding the characteristics and evolution of source-to-sink systems in rift settings globally and facilitates a deeper examination of fault–sediment system interactions under multi-stage extensional regimes. Such studies not only fill the existing research gap in S2S analysis within this basin but also contribute meaningful insights for hydrocarbon exploration and tectono-sedimentary evolution in other rift basins.

Through integrated geological, geophysical, and sedimentological analyses, this study further elucidates the spatiotemporal distrib

---

## [Decision Letter · Decision Letter 1]

26 Dec 2025

PONE-D-25-52722R1Study on Source- to-Sink System Coupling and Spatiotemporal Evolution of Multiphase Rift Basins: A Case Study of Jurassic-Cretaceous in Lishu Fault Depression, Southeastern Songliao BasinPLOS One

Dear Dr. Wang,

Thank you for submitting your manuscript to PLOS ONE. After careful consideration, we feel that it has merit but does not fully meet PLOS ONE’s publication criteria as it currently stands. Therefore, we invite you to submit a revised version of the manuscript that addresses the points raised during the review process.

If applicable, we recommend that you deposit your laboratory protocols in protocols.io to enhance the reproducibility of your results. Protocols.io assigns your protocol its own identifier (DOI) so that it can be cited independently in the future. For instructions see: https://journals.plos.org/plosone/s/submission-guidelines#loc-laboratory-protocols. Additionally, PLOS ONE offers an option for publishing peer-reviewed Lab Protocol articles, which describe protocols hosted on protocols.io. Read more information on sharing protocols at . Additionally, PLOS ONE offers an option for publishing peer-reviewed Lab Protocol articles, which describe protocols hosted on protocols.io. Read more information on sharing protocols at https://plos.org/protocols?utm_medium=editorial-email&utm_source=authorletters&utm_campaign=protocols..

We look forward to receiving your revised manuscript.

Kind regards,

Santanu Banerjee

Academic Editor

PLOS One

Journal Requirements:

Additional Editor Comments :

Comments to the Author

1. Is the manuscript technically sound, and do the data support the conclusions?

Reviewer #1: Partly

2. Has the statistical analysis been performed appropriately and rigorously?

Reviewer #1: NA

4. Have the authors made all data underlying the findings in their manuscript fully available?

Reviewer #1: Yes

5. Is the manuscript presented in an intelligible fashion and written in standard English?

Reviewer #1: No

6. Review Comments to the Author

Reviewer #1:

Overall Assessment

This manuscript presents a comprehensive study on the source-to-sink systems within the Lishu Fault Depression, utilizing seismic, well-log, and sedimentological data. The research is well-structured, methodologically sound, and provides valuable insights into the coupling mechanisms and spatiotemporal evolution of sedimentary systems in a multiphase rift basin. The findings have practical implications for hydrocarbon exploration in similar geological settings. However, several areas require clarification, refinement, and deeper discussion before the manuscript is suitable for publication.

（1） The current title is overly long, and beginning with "Study on" is redundant. It is recommended to revise it to a more concise and idiomatic expression. Some terms are also inaccurate.

（2） The abstract is information-dense yet overly condensed. Consider streamlining the content to more clearly highlight the key objectives, methods, findings, and implications. The transition between the results (①–⑥) is somewhat abrupt. A more coherent narrative would improve readability.

（3） The literature review in the introduction is comprehensive but appears lengthy and unfocused. It could be more sharply focused on recent advances in source-to-sink modeling within rift basins.

（4） The methodological description is detailed but could benefit from a clearer step-by-step explanation, particularly regarding the integration of different data types (seismic, logging, core).

（5） The discussion section touches on topics like global rift basin comparisons and quantitative relationships, which feels somewhat fragmented and lacks a tight connection to the core issue raised in the introduction: "the coupling mechanisms and evolutionary models of the source-to-sink system in the Lishu Fault Depression." It is recommended to streamline or integrate sections 5.3 and 5.4, emphasizing discussions directly relevant to this case study.

（6） The questions raised in the introduction, such as "how episodic rifting influences sediment stacking patterns" and "how to establish a source-to-sink coupling model," are addressed in the results and discussion, but the presentation could be more explicit. It is suggested to provide an itemized summary at the beginning of the discussion or in the conclusion section to directly respond to the scientific questions posed in the introduction.

（7） The conclusion section largely repeats the results and lacks a concise synthesis of the "coupling mechanisms" and "evolutionary models."

（8） Some figures lack complete information or have unclear labeling.

（9） The manuscript contains noticeable grammatical errors, unnatural expressions, and inaccurate or inconsistent terminology. A thorough language polish by a native English speaker or a professional editing service is highly recommended.

Summary of review: Although there are enough data, the paper needs to be organized properly. The authors should resubmit the paper if all comments of the reviewer are properly addressed and the paper is revised approriately

Reviewers' comments:

Reviewer's Responses to Questions

**Comments to the Author**

1. If the authors have adequately addressed your comments raised in a previous round of review and you feel that this manuscript is now acceptable for publication, you may indicate that here to bypass the “Comments to the Author” section, enter your conflict of interest statement in the “Confidential to Editor” section, and submit your "Accept" recommendation.

Reviewer #3: (No Response)

2. Is the manuscript technically sound, and do the data support the conclusions?

Reviewer #3: Partly

3. Has the statistical analysis been performed appropriately and rigorously? 

Reviewer #3: N/A

4. Have the authors made all data underlying the findings in their manuscript fully available?

Reviewer #3: Yes

5. Is the manuscript presented in an intelligible fashion and written in standard English?

Reviewer #3: No

6. Review Comments to the Author

Reviewer #3: Overall Assessment

This manuscript presents a comprehensive study on the source-to-sink systems within the Lishu Fault Depression, utilizing seismic, well-log, and sedimentological data. The research is well-structured, methodologically sound, and provides valuable insights into the coupling mechanisms and spatiotemporal evolution of sedimentary systems in a multiphase rift basin. The findings have practical implications for hydrocarbon exploration in similar geological settings. However, several areas require clarification, refinement, and deeper discussion before the manuscript is suitable for publication.

（1） The current title is overly long, and beginning with "Study on" is redundant. It is recommended to revise it to a more concise and idiomatic expression. Some terms are also inaccurate.

（2） The abstract is information-dense yet overly condensed. Consider streamlining the content to more clearly highlight the key objectives, methods, findings, and implications. The transition between the results (①–⑥) is somewhat abrupt. A more coherent narrative would improve readability.

（3） The literature review in the introduction is comprehensive but appears lengthy and unfocused. It could be more sharply focused on recent advances in source-to-sink modeling within rift basins.

（4） The methodological description is detailed but could benefit from a clearer step-by-step explanation, particularly regarding the integration of different data types (seismic, logging, core).

（5） The discussion section touches on topics like global rift basin comparisons and quantitative relationships, which feels somewhat fragmented and lacks a tight connection to the core issue raised in the introduction: "the coupling mechanisms and evolutionary models of the source-to-sink system in the Lishu Fault Depression." It is recommended to streamline or integrate sections 5.3 and 5.4, emphasizing discussions directly relevant to this case study.

（6） The questions raised in the introduction, such as "how episodic rifting influences sediment stacking patterns" and "how to establish a source-to-sink coupling model," are addressed in the results and discussion, but the presentation could be more explicit. It is suggested to provide an itemized summary at the beginning of the discussion or in the conclusion section to directly respond to the scientific questions posed in the introduction.

（7） The conclusion section largely repeats the results and lacks a concise synthesis of the "coupling mechanisms" and "evolutionary models."

（8） Some figures lack complete information or have unclear labeling.

（9） The manuscript contains noticeable grammatical errors, unnatural expressions, and inaccurate or inconsistent terminology. A thorough language polish by a native English speaker or a professional editing service is highly recommended.

7. PLOS authors have the option to publish the peer review history of their article (what does this mean?). If published, this will include your full peer review and any attached files.). If published, this will include your full peer review and any attached files.

.

Reviewer #3: No

---

## [Author Response · Author response to Decision Letter 2]

26 Jan 2026

Response to Reviewers

We sincerely appreciate the reviewers for their thorough evaluation of our manuscript and for providing insightful comments that greatly assisted us in improving the quality of the paper. We have carefully addressed each comment, and our detailed responses are provided directly below each corresponding remark in red text.Once again, we would like to express our gratitude for your time and constructive review of our work.

（1）The current title is overly long, and beginning with "Study on" is redundant. It is recommended to revise it to a more concise and idiomatic expression. Some terms are also inaccurate.

We thank the reviewer for this constructive and helpful suggestion. The title has been revised to be more concise and idiomatic by removing the redundant phrase “Study on” and by refining several terms. In particular, the expression “multiphase rift basins” has been replaced with “multiphase rifting” to more accurately describe the geological process, and the terminology of the source-to-sink system has been standardized.

The revised title now reads:

“Source-to-Sink Coupling and Spatiotemporal Evolution during Multiphase Rifting: The Jurassic-Cretaceous Lishu Fault Depression, Southeastern Songliao Basin.”

（2） The abstract is information-dense yet overly condensed. Consider streamlining the content to more clearly highlight the key objectives, methods, findings, and implications. The transition between the results (①–⑥) is somewhat abrupt. A more coherent narrative would improve readability.

We thank the reviewer for this constructive and insightful comment. In response, the abstract has been substantially revised to improve clarity, readability, and narrative coherence. Specifically, the content has been streamlined to more clearly emphasize the research objectives, data and methods, key findings, and geological implications. The previously itemized and condensed results have been reorganized into a continuous narrative with smoother transitions, thereby enhancing the logical flow of the abstract. These revisions improve the accessibility of the abstract while retaining the essential scientific contributions of the study.

Location in manuscript: Revised Abstract, Lines 21–46.

（3） The literature review in the introduction is comprehensive but appears lengthy and unfocused. It could be more sharply focused on recent advances in source-to-sink modeling within rift basins.

We thank the reviewer for this valuable suggestion. In response, we have revised the introduction to focus more sharply on recent advances in source-to-sink (S2S) modeling in rift basins. The revised text now emphasizes:

1) Quantitative and process-based S2S modeling approaches, including forward stratigraphic modeling, high-resolution seismic analysis, well-log interpretation, and provenance data.

2) How multiphase fault growth, drainage reorganization, and accommodation variations control sediment routing, depositional system development, and basin-scale stratigraphy.

3) Recent case studies that demonstrate the application of these methods to lacustrine and continental rift basins, highlighting their relevance to predicting sand-body distribution and reservoir potential.

Location in manuscript: Revised Introduction, Lines 51–83.

（4） The methodological description is detailed but could benefit from a clearer step-by-step explanation, particularly regarding the integration of different data types (seismic, logging, core).

We thank the reviewer for this constructive comment. In response, we have reorganized and revised the Methods section to provide a clear, step-by-step description of our workflow, explicitly highlighting the integration of seismic, well-log, and core data.

Specifically:

Stepwise workflow: The methodology is now presented in five sequential steps: fault activity quantification, paleogeomorphology reconstruction, provenance analysis, transport channel identification, and statistical analysis of sedimentary fans.

Data integration: At each step, we explicitly indicate how seismic, well-log, and core/petrographic data are combined to ensure consistency and improve interpretive robustness.

Quantitative analysis: Fault activity rates, valley geometry, and sedimentary fan statistics are calculated based on integrated datasets, and potential uncertainties (e.g., depth conversion and seismic-attribute extraction errors) are addressed.

Visualization: Paleogeomorphology and source-to-sink systems are now reconstructed and rendered in 3D to clearly demonstrate spatial relationships.

These revisions improve clarity, reproducibility, and readability, ensuring that the integration of multiple data types and the overall methodological workflow are transparent.

Location in manuscript: Revised Methods section, Lines 126–190.

（5） The discussion section touches on topics like global rift basin comparisons and quantitative relationships, which feels somewhat fragmented and lacks a tight connection to the core issue raised in the introduction: "the coupling mechanisms and evolutionary models of the source-to-sink system in the Lishu Fault Depression." It is recommended to streamline or integrate sections 5.3 and 5.4, emphasizing discussions directly relevant to this case study.

We appreciate the reviewer’s insightful comment. In response, we have revised Sections 5.3 and 5.4 to streamline the discussion and improve its focus. Specifically:

Integration and simplification: Global rift basin comparisons and quantitative relationships are now summarized concisely and only included where they directly inform the understanding of source-to-sink system evolution in the Lishu Fault Depression.

Enhanced linkage to core research question: All discussion points are now explicitly connected to the coupling mechanisms and evolutionary model of the Lishu Fault Depression S2S system, emphasizing fault activity, sediment routing, and depositional system response under multiphase rifting.

Improved narrative flow: The discussion now progresses logically from regional observations to case-specific interpretations, ensuring that broader examples support, rather than distract from, the core findings.

These revisions enhance coherence, focus, and relevance to the primary objectives of the study.

Location in manuscript: Revised Discussion, Lines 601–693.

（6） The questions raised in the introduction, such as "how episodic rifting influences sediment stacking patterns" and "how to establish a source-to-sink coupling model," are addressed in the results and discussion, but the presentation could be more explicit. It is suggested to provide an itemized summary at the beginning of the discussion or in the conclusion section to directly respond to the scientific questions posed in the introduction.

We thank the reviewer for this valuable suggestion. In response, we have revised the Discussion and Conclusions to provide a concise and explicit summary of how our findings address the key questions raised in the Introduction. Specifically:

Episodic rifting and sediment accumulation: The Discussion now clearly summarizes how variations in fault activity, drainage reorganization, and accommodation space during multiphase rifting controlled sediment routing, depositional system development, and sedimentary body scaling.

Source-to-sink coupling model: We explicitly highlight the reconstructed quantitative relationships among sediment supply, transport pathways, and depositional systems, providing a clear framework for a coupled S2S model of the Lishu Fault Depression.

Enhanced narrative linkage: These summaries are now included at the beginning of the Discussion and in the Conclusions, ensuring that the manuscript explicitly closes the loop between the research questions posed in the Introduction and the study results.

These revisions improve clarity and directly demonstrate how the study addresses the scientific objectives.

Location in manuscript: Revised Discussion, Lines 602–617.

（7） The conclusion section largely repeats the results and lacks a concise synthesis of the "coupling mechanisms" and "evolutionary models."

We thank the reviewer for this valuable suggestion. In response, we have substantially revised the Conclusion section to emphasize the coupling mechanisms and evolutionary model of the source-to-sink (S2S) system in the Lishu Fault Depression.

Specifically:

Integration of results into mechanisms: Rather than listing results, the revised conclusion now clearly links fault activity, sediment supply, and depositional processes, highlighting how differential fault activity controls accommodation space, sediment routing, and sand-body distribution.

Basin evolutionary model: We summarize the basin evolution from early rifting through rifting–contraction to shallow-water expansive systems, emphasizing how these stages are controlled by the interaction of tectonics and sediment transport.

Quantitative S2S coupling: Statistical relationships between geomorphic parameters (valley width, depth, cross-sectional area) and depositional system scale are now explicitly presented, demonstrating measurable links between tectonics, transport, and deposition.

These revisions provide a concise synthesis of both the mechanistic controls and the evolutionary framework, directly addressing the scientific objectives raised in the Introduction.

Location in manuscript: Revised Conclusions, Lines 740–763.

（8） Some figures lack complete information or have unclear labeling.

We thank the reviewer for this helpful comment. In response, we have carefully revised all figures to improve clarity and completeness:

1) Figure labeling: All figures now include fully labeled axes, units, legends, and annotations to clearly indicate key features such as faults, sequences, and depositional systems.

2) Figure captions: Captions have been expanded to provide concise descriptions of the data, interpretations, and symbols used in each figure.

3) Supplementary details: Where necessary, additional insets or enlarged views have been added to highlight critical features that were previously difficult to discern.

4) Overall clarity: Color schemes, line styles, and symbol sizes have been adjusted to ensure that all figures are legible in print and on-screen.

These revisions ensure that all figures are self-contained, clearly labeled, and fully support the interpretations presented in the text.

（9） The manuscript contains noticeable grammatical errors, unnatural expressions, and inaccurate or inconsistent terminology. A thorough language polish by a native English speaker or a professional editing service is highly recommended.

Thank you very much for this constructive comment. We fully acknowledge that the previous version of the manuscript contained grammatical errors, awkward expressions, and some inconsistencies in terminology.

In response to this concern, the entire manuscript has been thoroughly revised and professionally edited by a native English speaker with experience in scientific writing. During this process, we corrected grammatical and syntactic errors, improved sentence fluency and readability, and carefully standardized technical terms to ensure accuracy and consistency throughout the manuscript.

We believe that these revisions have significantly improved the overall language quality and clarity of the manuscript, making it more suitable for publication. We sincerely appreciate the reviewer’s suggestion, which has helped us to substantially enhance the presentation of our work.

---

## [Decision Letter · Decision Letter 2]

7 Apr 2026

PONE-D-25-52722R2Source-to-Sink Coupling and Spatiotemporal Evolution during Multiphase Rifting: The Jurassic-Cretaceous Lishu Fault Depression, Southeastern Songliao BasinPLOS One

Dear Dr. Wang,

Thank you for submitting your manuscript to PLOS ONE. After careful consideration, we feel that it has merit but does not fully meet PLOS ONE’s publication criteria as it currently stands. Therefore, we invite you to submit a revised version of the manuscript that addresses the points raised during the review process.

If applicable, we recommend that you deposit your laboratory protocols in protocols.io to enhance the reproducibility of your results. Protocols.io assigns your protocol its own identifier (DOI) so that it can be cited independently in the future. For instructions see: https://journals.plos.org/plosone/s/submission-guidelines#loc-laboratory-protocols. Additionally, PLOS ONE offers an option for publishing peer-reviewed Lab Protocol articles, which describe protocols hosted on protocols.io. Read more information on sharing protocols at . Additionally, PLOS ONE offers an option for publishing peer-reviewed Lab Protocol articles, which describe protocols hosted on protocols.io. Read more information on sharing protocols at https://plos.org/protocols?utm_medium=editorial-email&utm_source=authorletters&utm_campaign=protocols..

As the corresponding author, your ORCID iD is verified in the submission system and will appear in the published article. PLOS supports the use of ORCID, and we encourage all coauthors to register for an ORCID iD and use it as well. Please encourage your coauthors to verify their ORCID iD within the submission system before final acceptance, as unverified ORCID iDs will not appear in the published article. *Only* the individual author can complete the verification step; PLOS staff the individual author can complete the verification step; PLOS staff *cannot* verify ORCID iDs on behalf of authors.verify ORCID iDs on behalf of authors.

We look forward to receiving your revised manuscript.

Kind regards,

Santanu Banerjee

Academic Editor

PLOS One

Journal Requirements:

Additional Editor Comments:

The paper needs moderate revision as suggested by the reviewers. You need to carefully address the comments and revise the paper accordingly.

Reviewer 1 comments:

（1）The Abstract and Conclusion sections should be further refined and elevated to better summarize the distinct characteristics of each part of the study.

（2）It is suggested to clarify how segmented fault activity drives the migration of depocenters, as well as the differentiation patterns of sedimentary facies between steep and gentle slope zones.

（3）The source system analysis could be further streamlined. Focus on elucidating the evolutionary partitioning of source and ensure that the source evolution across different rifting stages is clearly articulated.

（4）The Discussion should more explicitly define the coupling mechanisms between these three components.

（5）Please remove unused well locations from the figures (e.g., Figure 1, Figure 16, etc.).

（6）A scale bar should be added to Figure 15.

（7）In Figure 17 (the conceptual model), briefly indicate the specific types of S2S systems developed within various structural domains of the study area.

（8）It is suggested that the discussion in this paper delve deeper into these relationships to strengthen the quantitative aspect of the S2S research.

Reviewers' comments:

Reviewer's Responses to Questions

**Comments to the Author**

1. If the authors have adequately addressed your comments raised in a previous round of review and you feel that this manuscript is now acceptable for publication, you may indicate that here to bypass the “Comments to the Author” section, enter your conflict of interest statement in the “Confidential to Editor” section, and submit your "Accept" recommendation.

Reviewer #4: (No Response)

Reviewer #5: All comments have been addressed

2. Is the manuscript technically sound, and do the data support the conclusions?

Reviewer #4: Yes

Reviewer #5: Yes

3. Has the statistical analysis been performed appropriately and rigorously? 

Reviewer #4: Yes

Reviewer #5: Yes

4. Have the authors made all data underlying the findings in their manuscript fully available?

Reviewer #4: Yes

Reviewer #5: Yes

5. Is the manuscript presented in an intelligible fashion and written in standard English?

Reviewer #4: Yes

Reviewer #5: Yes

6. Review Comments to the Author

Reviewer #4: （1）The Abstract and Conclusion sections should be further refined and elevated to better summarize the distinct characteristics of each part of the study.

（2）It is suggested to clarify how segmented fault activity drives the migration of depocenters, as well as the differentiation patterns of sedimentary facies between steep and gentle slope zones.

（3）The source system analysis could be further streamlined. Focus on elucidating the evolutionary partitioning of source and ensure that the source evolution across different rifting stages is clearly articulated.

（4）The Discussion should more explicitly define the coupling mechanisms between these three components.

（5）Please remove unused well locations from the figures (e.g., Figure 1, Figure 16, etc.).

（6）A scale bar should be added to Figure 15.

（7）In Figure 17 (the conceptual model), briefly indicate the specific types of S2S systems developed within various structural domains of the study area.

（8）It is suggested that the discussion in this paper delve deeper into these relationships to strengthen the quantitative aspect of the S2S research.

Reviewer #5: The authors have carefully addressed all the previous reviewers' comments point by point and provided a detailed response. In my opinion, the current version of the manuscript has reached the quality standards required for publication in this journal. I recommend acceptance.

7. PLOS authors have the option to publish the peer review history of their article (what does this mean?). If published, this will include your full peer review and any attached files.). If published, this will include your full peer review and any attached files.

.

Reviewer #4: No

Reviewer #5: No

---

## [Author Response · Author response to Decision Letter 3]

13 Apr 2026

We sincerely appreciate the reviewers for their thorough evaluation of our manuscript and for providing insightful comments that greatly assisted us in improving the quality of the paper. We have carefully addressed each comment, and our detailed responses are provided directly below each corresponding remark in red text.Once again, we would like to express our gratitude for your time and constructive review of our work.

1. Overall Assessment: This study successfully reveals the coupling relationship between source evolution, fault activity, and sedimentary sand bodies, providing an important basis for rift basin evolution and hydrocarbon exploration. The Abstract and Conclusion sections should be further refined and elevated to better summarize the distinct characteristics of each part of the study.

Thank you for your comment. In response to your review suggestions, we have revised the Abstract and Conclusions sections accordingly and supplemented the description of source-to-sink system evolution types during the rifting stages. Please see Lines 40–45 in the Abstract and Lines 774–782 in the Conclusions for details.

The added content in the Abstract is as follows:“During the initial rifting stage, steep-slope nearshore subaqueous fans and slope–fan delta systems developed. In the early phase of intense rifting, dominant patterns included steep slope–transfer zone–fan delta, axial slope–fan delta, and eastern gentle slope parallel fault-step zone systems. In the late phase, axial drainage systems prevailed, forming axial slope–braided river delta patterns. During the late rifting stage, gentle slope–braided river delta source-to-sink systems became dominant.”

The added content in the Conclusions is as follows: “During the initial rifting stage, sediment supply was partitioned, leading to the development of steep-slope nearshore subaqueous fans and a differentiated source-to-sink coupling pattern of steep slope–fan delta systems. In the early stage of intense rifting, sediment supply became convergent, and the dominant source-to-sink coupling patterns included steep slope–transfer zone–fan delta systems, axial slope–fan delta systems, and eastern gentle slope parallel fault-step systems. In the late stage, axial drainage systems became dominant, primarily forming axial slope–braided river delta coupling systems. During the late rifting stage, axial sediment supply was further strengthened, and the system was characterized by the development of gentle slope–braided river delta source-to-sink coupling patterns.”

2. Tectonic-Sedimentary Coupling: It is suggested to clarify how segmented fault activity drives the migration of depocenters, as well as the differentiation patterns of sedimentary facies between steep and gentle slope zones.

Thank you for your valuable comments. In response to your suggestions, we have added a discussion on how segmented fault activity evolution drives the migration of the depocenter, which has been incorporated into the section on fault activity analysis (see Lines 238–265). In addition, we have supplemented the description of the differential distribution patterns of sedimentary facies between the steep-slope and gentle-slope zones (see Lines 629–684).

3. Source Analysis: The source system analysis could be further streamlined. Focus on elucidating the evolutionary partitioning of source and ensure that the source evolution across different rifting stages is clearly articulated.

Thank you for your valuable comments. In the provenance system analysis section, we have summarized the sediment sources of different rifting phases, and the content has been further streamlined. We can clearly identify the changes in provenance for each rifting phase. Please refer to Lines 308–353 for details.

4. Discussion Section: The paper provides a detailed summary of the S2S evolution in the Lishu Fault Depression, emphasizing the coupling of "source system-transport pathway-depositional system." The Discussion should more explicitly define the coupling mechanisms between these three components.

Thank you for your valuable comments. In response to your suggestions, we have added a description of the coupling relationships among the provenance system, transport pathways, and depositional system in the Discussion section. Please see Lines 636–684 for details.

5. Figures: Please remove unused well locations from the figures (e.g., Fig 1, Fig 16, etc.).

Thank you for your valuable comments. In response to your suggestions, we have removed unnecessary well locations from Fig 1, Fig 16, and other relevant figures.

6. Scale Bars: A scale bar should be added to Figure 15.

Thank you for your comment. In response to your review, we have added a scale bar to Fig 15.

7. Schematic Model: In Figure 17 (the conceptual model), briefly indicate the specific types of S2S systems developed within various structural domains of the study area.

Thank you for your valuable comments. In response to your suggestions, we have revised Fig 17 to include the types of source–sink systems developed in different structural domains.

8. Quantitative Analysis: Regarding Table 2, previous studies have frequently discussed the relationship between paleo-valley parameters in the source area (width, depth, width-to-depth ratio, cross-sectional area) and the scale of corresponding fans in the depositional area. It is suggested that the discussion in this paper delve deeper into these relationships to strengthen the quantitative aspect of the S2S research.

We sincerely thank you for your comments. In response to your suggestions, we have added a discussion on the quantitative analysis of the source-to-sink system in the Discussion section. Please refer to lines 744–753 for the detailed revisions.

The detailed content is as follows: “Overall, the scale of depositional areas in multi-stage rift sedimentary systems is closely related to the characteristics of transport pathways, and is also partly controlled by the rift evolutionary stage. During the early rifting stage, deformation is dominated by lateral extension, and basin stretching is mainly controlled by fault activity, resulting in proximal deposition. Under conditions of stronger fault activity, the areal extent of fan systems tends to be relatively small. In contrast, during the intense rifting stage, sediment supply is sufficient and fault activity is further enhanced, leading to a significant increase in the axial extent of depositional systems. Therefore, the sedimentary area within source-to-sink systems is to a large extent governed by the tectonic phases of the rift basin, while fault activity intensity and stress orientation are the key factors controlling the architecture of the source-to-sink system.”

---

## [Editor Report · Decision Letter 3]

14 Apr 2026

Source-to-Sink Coupling and Spatiotemporal Evolution during Multiphase Rifting: The Jurassic-Cretaceous Lishu Fault Depression, Southeastern Songliao Basin

PONE-D-25-52722R3

Dear Dr. Wang,

We’re pleased to inform you that your manuscript has been judged scientifically suitable for publication and will be formally accepted for publication once it meets all outstanding technical requirements.

An invoice will be generated when your article is formally accepted. Please note, if your institution has a publishing partnership with PLOS and your article meets the relevant criteria, all or part of your publication costs will be covered. Please make sure your user information is up-to-date by logging into Editorial Manager at Editorial Manager® and clicking the ‘Update My Information' link at the top of the page. For questions related to billing, please contact  and clicking the ‘Update My Information' link at the top of the page. For questions related to billing, please contact billing support..

Kind regards,

Santanu Banerjee

Academic Editor

PLOS One

Additional Editor Comments (optional):

The paper has been revised appropriately.
---

## [Editor Report · Acceptance letter]

PONE-D-25-52722R3

PLOS One

Dear Dr. Wang,

I'm pleased to inform you that your manuscript has been deemed suitable for publication in PLOS One. Congratulations! Your manuscript is now being handed over to our production team.

Kind regards,

on behalf of

Dr. Santanu Banerjee

Academic Editor

PLOS One